# A data-driven evaluation of post-fire landslide susceptibility

Elsa S. Culler[1, 2], Ben Livneh[1, 2], Balaji Rajagopalan[1], and Kristy F. Tiampo[3, 2]

[1]University of Colorado Boulder Department of Civil, Architectural, and Environmental Engineering
[2]Cooperative Institute for Research in Environmental Sciences (CIRES), University of Colorado Boulder
[3]University of Colorado Boulder Department of Geologic Sciences

**Correspondence:** Elsa Culler (elsa.culler@colorado.edu)

**Abstract.** Wildfires change the hydrologic and geomorphic response of watersheds, which has been associated with cascades of additional hazards and management challenges. Among these post-wildfire events are shallow landslides and debris flows. This study evaluates post-wildfire mass movement trigger characteristics by comparing the precipitation preceding events at both burned and unburned locations. Landslide events are selected from the NASA Global Landslide Catalog (GLC). Since this catalog contains events from multiple regions worldwide, it allows a greater degree of inter-region comparison than many more localized catalogs. Fire and precipitation histories for each site are established using MODIS burned area and CHIRPS precipitation data, respectively. Analysis of normalized seven-day accumulated precipitation for sites across all regions shows that, globally, mass movements at burned sites are preceded by less precipitation than mass movements without antecedent burn events. This supports the hypothesis that fire increases rainfall-driven mass movement hazards. An analysis of the seasonality of mass movements at burned and unburned locations shows that mass movement-triggering storms in burned locations tend to exhibit different seasonality from rainfall-triggered mass movements in areas undisturbed by recent fire, with a variety of seasonal shifts ranging from approximately six months in the Pacific Northwest of North America to one week in the Himalaya region. Overall, this manuscript offers an exploration of regional differences in the characteristics of rainfall-triggered mass movements at burned and unburned sites over a broad spatial scale and encompassing a variety of climates and geographies.

## 1 Introduction

Mass movements are destructive when they occur near vulnerable areas, causing damage to buildings, utility lines, and roadways (Highland and Bobrowsky, 2008). Landslide mitigation costs in the United States (US) are approximately 2 billion USD annually, with worldwide costs much higher (Schuster and Highland, 2001). There can also be indirect impacts, such as aggradation of the streambed, or the formation of landslide dams (Glade and Crozier, 2005). Worldwide, these natural disasters cause tens of thousands of deaths each year (Froude and Petley, 2018). Landslide mitigation costs in the United States (US) are approximately 2 billion USD annually, with worldwide costs much higher (Schuster and Highland, 2001). Though an accurate assessment of mass movement hazards would aid mitigation efforts (Spiker and Gori, 2002), such an evaluation presents a challenge in part because mass movements are often triggered by a sequence of cascading natural hazards (Klose, 2015a). For example, mass movements may interact with other complex phenomena such as heavy rain, wildfires, floods, earthquakes,

melting permafrost and glacial outbursts (Budimir et al., 2015; Harp et al., 2011; Kirschbaum et al., 2020, 2012; Rupert et al., 2003).

Here, we focus on a particular sequence of cascading natural hazards known as the post-wildfire landslide. In these events, wildfires are followed by intense precipitation leading to mass movements such as shallow landslides, or debris flows. The impact of wildfires, which themselves occur more frequently and severely as a consequence of higher temperatures and increasingly widespread drought, can lead to a variety of geo-hydrological hazards including increased snowmelt, water contamination, increased erosion rates, and decreased infiltration (AghaKouchak et al., 2020). Post-wildfire landslides in particular occur when wildfires are followed by intense precipitation, leading to mass movements such as a sediment-laden floods, shallow landslides, or debris flows. We use the term 'mass movement' preferentially over landslide in this study in order the encompass this variety of phenomena, not all of which are landslides in the strictest definition. Nonetheless, when describing prior literature in which 'landslides' and 'mass movements' are used interchangeably we defer to the terminology used by the authors of the cited work.

The impact of wildfire on landslide hazards can vary on the basis of static factors such as burn severity, vegetation, and soil types (Cannon et al., 2010; Staley et al., 2018). Mass movement hazards in general may also depend on dynamic factors such as soil moisture, meteorology and the length of time since the most recent fire (Kirschbaum and Stanley, 2018; DeGraff et al., 2015; McGuire et al., 2021). There are numerous local studies demonstrating a relationship between wildfire occurrence or severity and the amount of precipitation that triggers a mass movement (Cannon et al., 2008; Gartner, 2005; Reneau et al., 2007; Riley et al., 2013) . The impact of wildfire on landslide hazards can also vary on the basis of local factors such as vegetation, and soil type (Cannon et al., 2010; Staley et al., 2018). In general, the lack of complete landslide inventories including a wide variety of climates and ecoregions presents an obstacle to evaluating the role of fire in rainfall-triggered landslides(Klose, 2015b).

This study seeks to test the hypothesis that wildfire consistently increases mass movement susceptibility across six global regions by detecting and characterizing differences in mass movement-triggering precipitation at both burned and unburned sites. Though we cannot draw conclusions about the susceptibility leading to any particular event, less precipitation among a group of post-wildfire suggests that the threshold for triggering a mass movement was lowered, e.g. susceptibility was greater. A second purpose of this study is to explore the possibility that the relationship between wildfire history and the expected frequency of landslide-triggering precipitation varies by region.

The GLC provides a large collection of rainfall-triggered landslides taking place in a variety of climates such that, in combination with spatially continuous observations of fire (500m Moderate Resolution Imaging Spectroradiometer [MODIS] Burned Area by Giglio et al. (2018)) and precipitation (5.5km Climate Hazards group InfraRed Precipitation with Station data [CHIRPS] by Funk et al. (2015)) data, it is well suited for comparing the diverse precursors under which post-wildfire mass movements occur.

## 1.1 Mechanisms by which fire increases mass movement hazards

While many factors contribute to mass movement hazards, only a subset are altered by fire exposure (Highland and Bobrowsky, 2008), and are therefore of interest to this analysis. Fire changes hydrologic and geomorphic response through several distinct physical mechanisms. First, the destruction of vegetation contributes to the development of debris flows and other mass movements in three ways:

- Sediment gathered behind vegetation trunks and stems can, after a fire, be mobilized either by a rain storm or as dry ravel, i.e. sediment that rolls down the slope without precipitation (Cannon and Gartner, 2005).

- Vegetation destruction clears pathways for water and sediment to flow downhill more quickly (Shakesby and Doerr, 2006).

- Following a fire, canopy and litter storage - water that gets trapped in leaves and other detritus on the ground - is greatly reduced, resulting in increased runoff and sediment transport (Cannon and Gartner, 2005; Shakesby and Doerr, 2006).

Additionally, soil properties can be dramatically altered post-fire, resulting in the following changes which can affect the formation of mass movements:

- Burned soils can have reduced organic content as a result of the combustion process, which causes them to have reduced water-holding capacity (Neary et al., 2005).

- Combustion of organic content also typically reduces soil aggregate stability, promoting erosion (Shakesby and Doerr, 2006).

- Some combinations of soil, vegetation type, and temperature can decrease wettability or produce a hydrophobic layer 1-5 cm beneath the soil, thereby dramatically increasing runoff (Spittler, 1995). The implications of this effect vary from place to place, since fire can also destroy hydrophobic layers in the right conditions. In addition, these effects are not always uniform across the burned area, and the effects of changed wettability can last from days to years depending on the local conditions (Shakesby and Doerr, 2006).

- A layer of post-fire ash caused by fire can also increase soil storage potential depending upon the thickness and hydraulic conductivity of the layer (Ebel et al., 2012).

One consequence of wildfire-driven changes to soil and vegetation on rainfall-triggered mass movements is that the predominant mechanism shifts from infiltration-driven to runoff-driven (Cannon and Gartner, 2005). Infiltration-driven mass movements are typically shallow slope failures initiated by longer storms that saturate the shallow subsurface. By contrast, runoff-driven mass movements are often debris flows caused by high-volume storms that mobilize sediment on the surface without the need for much infiltration. Mass movements can often be identified as one type or another primarily by observing

whether it had a point origin, as with infiltration-driven mass movements, or a distributed origin like runoff-driven mass movements. For infiltration-driven mass movements, the antecedent soil moisture conditions are more important for evaluating mass movement hazards since soil saturation is fundamental to the mechanism of slope failure. However, post-wildfire mass movements tend to be less driven by infiltration since the hydrophobic and more erodible sediment layer creates an ideal condition for runoff-driven mass movements. (Cannon et al., 2008; Santi and Rengers, 2020; Parise and Cannon, 2012).

## 1.2 Evidence for increased mass movement hazards with increased burn severity

Wildfire has been empirically linked to increased frequency and volume of debris flows in several regions of the Western US (Cannon and Gartner, 2005). A key piece of evidence for this connection comes from a series of studies based on repeated post-storm observations of burned watersheds in Southern California and the Intermountain West regions of the US as part of the development of the US Geological Survey's (USGS) operational post-wildfire mass movement hazard predictions (Cannon et al., 2010; Gartner et al., 2009, 2014; Rupert et al., 2003; Staley et al., 2016). These five studies model the probability of mass movements following fire using logistic regressions to demonstrate that both burn severity (Staley et al., 2016) and burn extent within a watershed (Cannon et al., 2010) are associated with increased debris flow likelihood. Notably, burn severity and extent are both increased by drought and other low antecedent soil moisture (Westerling and Swetnam, 2003), and thus we expect to find more post-wildfire debris flows in dry climates. Gartner et al. (2014) found that the increase in debris flow probability in a watershed due to wildfire is greatest immediately after wildfire, but can last a total of 2-5 years. Other studies suggest that the overall mass movement hazard evolves over time in a more complex manner, with debris flow hazards increasing for the year after the fire followed by an increase in the frequency of shallow landslides as tree roots decay in subsequent years (Rengers et al., 2020; Benda and Dunne, 1997). Increased likelihood of post-wildfire debris flows has also been associated with the erodibility of fine sediment in the soil, soil organic matter percentage, soil clay percentage, underlying lithology (e.g. sedimentary or granitic rock), watershed area, and watershed relief ratio (Gartner et al., 2009; Rupert et al., 2003; Pelletier and Orem, 2014).

The widely recognized relationship between mass movements and burn severity suggests that mass movement susceptibility increases after wildfires in the Western US, although none of the above studies include observations of unburned sites as a control. Instead, the databases used in Cannon et al. (2010); Gartner et al. (2009, 2014); Rupert et al. (2003); Staley et al. (2016) include multiple observations of the presence or absence of a debris flow at each site, making them suitable for a regression analysis based on burn severity, but not for comparing burned and unburned locations. In addition, while these post-wildfire mass movement observations contain precise dates and locations, and extend across a remarkable spatial range when compared to most other mass movement hazard models, they still are limited to 119 sites or fewer (Gartner et al., 2014). This limited spatial extent leaves open the question of whether the fire-flood patterns of the Western US are unique, or if similar hazards are just as ubiquitous but under-reported in other regions. A global study by Riley et al. (2013) comparing post-wildfire a non-fire-related debris flows found that the volumes of the post-wildfire debris flows tended to be smaller. This finding suggests that the increase in debris flow hazard and frequency after wildfires occurs in a variety of environments.

## 1.3 Sources and methods for mass movement data collection

It is resource-prohibitive to conduct a continuous systematic search for mass movements either in the field or with satellite observations. As a result, many of the most accurate and complete methods for systematically identifying mass movements can presently only be used over limited spatial and temporal domains. For example, Lee and Pradhan (2007) identified landslides from aerial photograph interpretation and a field survey over the $\sim 800\,\mathrm{km}^2$ Selangor area in Malaysia, and Nefeslioglu et al. (2010), used an inventory based on aerial photographs taken in 1955-1956 to analyze landslide susceptibility over a $\sim 175\,\mathrm{km}^2$

area near Istanbul, Turkey. An alternative to manual identification either in the field or using photographs is automatic or semi-automatic landslide detection using image processing on aerial imagery, LiDAR surveys, or Synthetic Aperture Radar (SAR). These automated methods are typically applied over similarly small domains due to challenges with obtaining imagery and compiling training datasets. For example, Martha et al. (2013) used aerial imagery over $\sim 120\,\mathrm{km}^2$ in the Himalayas, while Mezaal et al. (2017) used LiDAR over the $26.7\,\mathrm{km}^2$ Cameron Highlands of Malaysia. SAR interferometry can be

used for identification of pre-landslide motion, as was done by Lu et al. (2012) over the $\sim 1500\,\mathrm{km}^2$ Arno basin in Italy. In addition, several SAR techniques have been employed to identify post-landslide scars, including SAR amplitude mapping of landslides triggered by the Gorkha, Nepal earthquake in 2015 over a $14,500\,\mathrm{km}^2$ area (Meena and Tavakkoli Piralilou, 2019), coherence mapping of interferometric SAR the same earthquake-triggered landslides (Burrows et al., 2019), and the wildfire-triggered landslides over $\sim 60\,\mathrm{km}^2$ of the area burned by the 2017 Thomas Fire in California (Donnellan et al., 2018).

While automated mass movement detection as deployed in the above studies is continually undergoing promising advances, at the time of this analysis it has not yet been used to compile an inventory over a broad enough spatial domain to facilitate inter-regional comparisons. Such records collected in an uncoordinated effort over small domains are unsuitable for regional inter-comparisons such as we have undertaken here because these records do not contain standardized information for every region, are often unpublished (van Westen et al., 2006), and are unlikely to have daily temporal resolution that would allow

comparison with the precipitation record (Kirschbaum et al., 2010).

For this study, we chose to use the NASA Global Landslide Catelog (GLC, Kirschbaum et al., 2010). As with the few other regional and global databases available, the broad domain of the GLC comes coupled with issues of location error and spatial bias. For each landslide location, the GLC contains an estimate of the area in which the landslide occurred, labeled the "location accuracy". For consistency, we refer to this parameter using the same name. The sources of GLC data are second-hand

observations made by the news media, governmental organizations such as departments of transportation, and some available scientific reports (Kirschbaum et al., 2010). The absence of a systematic search for mass movements across the entire database domain results in a substantial spatial bias towards populated areas where mass movements happen to be noticeable. News reports also suffer relatively high location uncertainty (as much as 50 km) depending on how specific the source article is about the location (Kirschbaum et al., 2010). Finally, though the GLC does contain some information about the mass movement

mechanisms that would allow mass movements to be classified, for example, as debris flows or shallow landslides, the majority of the events in the GLC are labeled as a non-specific 'landslide' type, which could refer to any type of mass movement. Despite limitations in accuracy and completeness, the GLC was chosen for this study primarily because as of this writing it

offers the largest spatial and temporal range of any catalog. The GLC contains a sample of mass movements from across the globe ($n = 11377, 5313$ of which met study requirements — see Section 2.1), and a substantial proportion of mass movements were identified in this study as having occurred in recently burned areas ($n = 489; 9.2\%$).

## 1.4 Towards a global picture of mass movement susceptibility

This study seeks to test the hypothesis that wildfires increase landslide susceptibility by evaluating antecedent precipitation at both burned and unburned mass movement locations. Some existing local and regional studies (Cannon et al., 2010; Rupert et al., 2003) have assessed the impact of wildfire on mass movement susceptibility, but have not included unburned locations in their analyses. Other studies have also featured the GLC data and a global spatial extent, with a focus on validating large-scale mass movement hazard models (Kirschbaum and Stanley, 2018). This analysis is unique from other regional and global studies in that it combines the broad scope of the GLC data with an exploration of the role of wildfire in mass movement susceptibility. This study is also distinct from others that focus on the role of wildfire on mass movement sites (Gartner et al., 2009) in that here burned sites are contrasted with unburned sites instead of previous observations of the same location. Finally, in contrast to post-wildfire mass movement studies focused on a specific regions like the western US (Cannon and DeGraff, 2009), southern California (Gartner et al., 2014), Western Canada (Jordan, 2015), Korea (Lee et al., 2019) or southeast Australia (Nyman et al., 2011), this study combines the GLC with globally-observed fire and precipitation data to offer unique insights into the role of fire on mass movement susceptibility in diverse regions across the globe.

## 2 Methods

We first describe the mass movement data (Sect. 2.1), the study regions (Sect. 2.2) and fire data (Sect. 2.3). Mass movements were included only if precipitation data and at least 3 years antecedent fire data were available. The mass movements occurred between 2007 and 2019, with corresponding precipitation and fire data extending as far as 2004-2019 so as to capture antecedent conditions. The precipitation data (Sect. 2.4) leading up to the date of each mass movement were compared using three approaches. First, the seven-day running total precipitation depth percentile for the 30 days surrounding the day of the year and across the total 38-year record (see Sect. 2.4) was used as a proxy for mass movement susceptibility. We assume here that greater susceptibility results in a lower precipitation threshold to trigger a landslide. An observation, therefore, of lower precipitation percentile values triggering mass movements across a sample of sites suggests that susceptibility is generally higher in that group. This principle is illustrated in the susceptibility-based rainfall threshold model developed by (Monsieurs et al., 2019), in which the predicted threshold of antecedent rainfall resulting in a landslide is adjusted according to susceptibility factors. This percentile value was compared between burned and unburned sites within each region and for all included mass movements (see Sect. 2.5). Next, seven-day precipitation percentiles were compared with bootstrapped samples from burned and unburned sites separately (see Sect. 2.6) to confirm the findings from the depth percentile analysis and also to draw out differences in storm timing between burned and unburned groups. Finally, the precipitation frequency in the burned and unburned groups in the months and years surrounding each mass movement (see Sect. 2.7) was examined to identify shifts in

the seasonality of mass movements at burned sites relative to the unburned group. These seasonality results were augmented with kernel density estimates of mass movement occurrence by day-of-year at burned and unburned sites for each region.

## 2.1 Mass movement data

A sample ($n = 5313$) of rainfall-triggered mass movements was obtained from the GLC. Mass movement locations are shown in Fig. 1, along with a summary of fire and precipitation information obtained for those locations from the sources listed in

Table 1 (see Sects. 2.3 and 2.4). The GLC provides a large collection of events taking place in a variety of climates such that, in combination with spatially continuous observations of fire (500m Moderate Resolution Imaging Spectroradiometer [MODIS] Burned Area by Giglio et al., 2018) and precipitation (5.5km Climate Hazards group InfraRed Precipitation with Station data [CHIRPS] by Funk et al., 2015) data, it is well suited for comparing the diverse precursors under which post-wildfire mass movements occur.

In order to reduce errors resulting from including a variety of types of rainfall-triggered mass movements within the same dataset, the selected mass movements were limited to those labeled in the GLC with a 'landslide trigger' value of 'rain,' 'downpour,' 'flooding,' or 'continuous rain.' Mass movements with a second trigger such as an earthquake were eliminated. Snowmelt-driven mass movements were also not included because the impact of precipitation can be delayed in those cases. An analysis of the snow record in California/Nevada revealed only a single event with enough antecedent snow to suggest it

could have been mislabeled. Only records with location uncertainty of 10 km or less were included, since the mass movements with lower location accuracy presented problems for wildfire classification. Finally, only mass movements between $50°$S and $50°$N latitude were included, and the events occurring before the year 2000 were omitted, so as to ensure coverage by both fire and precipitation datasets (see Table 1).

The GLC contains a variety of types of rainfall-triggered mass movements with different physical mechanisms, including

205 debris flows, shallow landslides, and rock falls. The majority of included mass movements (65 %), however, are categorized simply as 'landslide', which according to the dataset authors can mean any type of mass movement. Since most of the mass movements are of an unknown type, we did not exclude data on the basis of category. Of the specific types of mass movements, most are labeled mudslides (25 %), with the next largest category being rockfalls at 4 %. This uncertainty as to landslide mechanism is currently a necessary trade-off for large spatial scales. This limitation highlights the need for large-scale catalogs

for specific types of mass movements, such as debris flows or shallow landslides.

## 2.2 Study regions

To compare the differences in mass movement triggers in different climates, we divided the mass movements into regions (see Fig. 1 panels (a)and (b)). Regions were determined using the AGglomerative NESting (AGNES) hierarchical clustering algorithm (Kaufman and Rousseeuw, 2009) considering the latitude and longitude of the mass movements, and clusters were

215 subsequently combined, split, or eliminated on the basis of equalizing sample sizes as described below. Though the regions are still large enough to encompass considerable variability in climate, the spatial clustering helps to ensure that the variability across regions - particularly in latitude - is larger than the variability within.

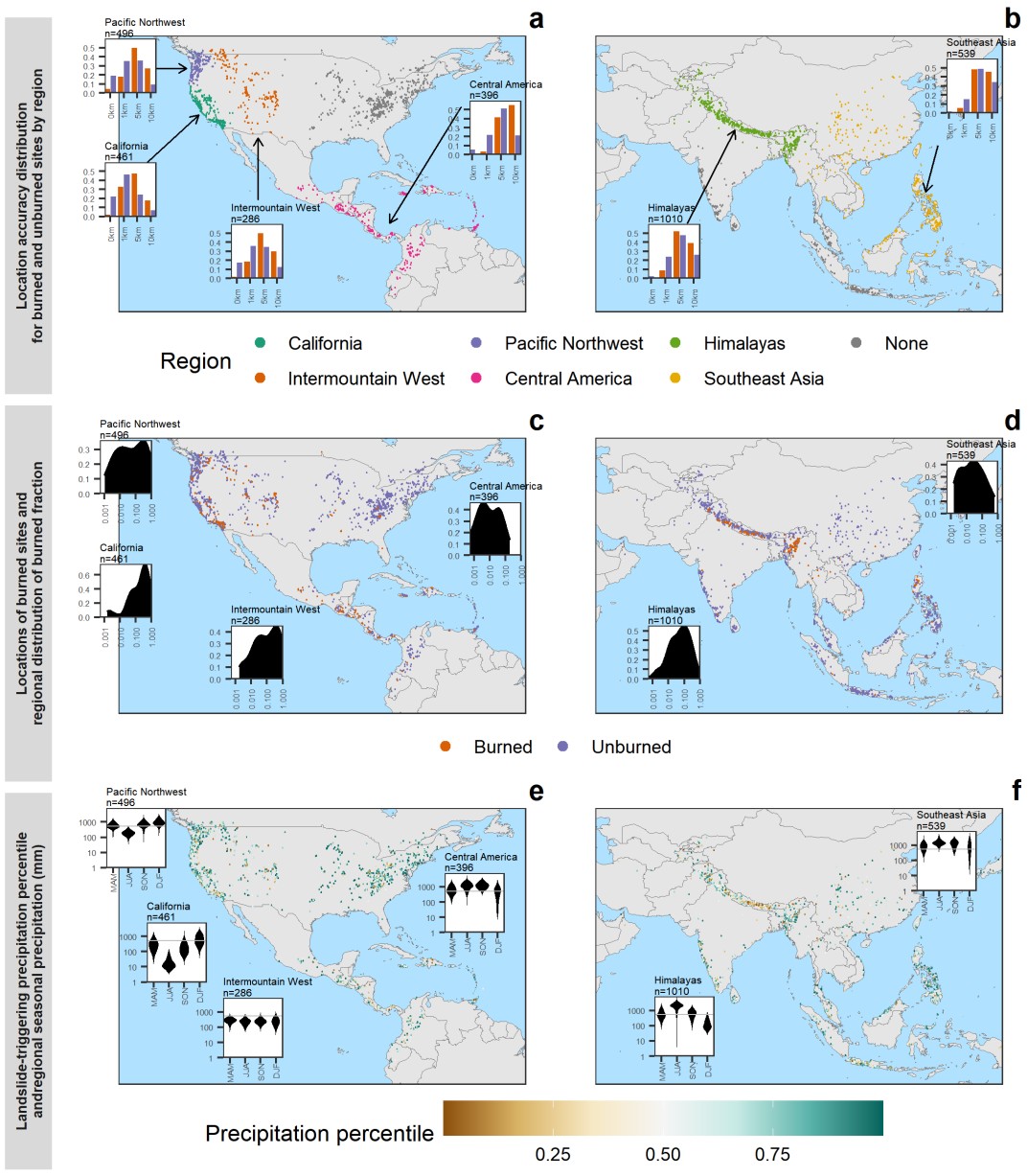

**Figure 1.** Landslide locations ($n = 5313$, $2006 - 2017$), showing region coding (see Sect. 2.2) in (a) and (b), with location accuracy for burned and unburned groups in the regional insets; burned/unburned classification at the time of the mass movement in (c) and (d), with regional insets showing kernel density portrayal of the fraction of burned area for the mass movement locations from the three years preceding the mass movement; and the precipitation percentile on the day of the mass movement in (e) and (f), with regional insets of kernel density estimates (violin plots) of the climatological ($1981 - 2020$) seasonal precipitation magnitude (mm) including a reference line indicating the median seasonal average across all sites globally. Country boundaries were obtained from the maps R package (Deckmyn et al., 2018)

**Table 1.** Description of datasets used in the analysis

| Data source | Description | Spatial extent | Spatial Resolution | Temporal Range | Temporal Resolution |
|---|---|---|---|---|---|
| NASA Global Landslide Catalog (GLC; Kirschbaum et al., 2010) | Compilation of landslides drawn from news articles and scientific reports | Global, with variable coverage in different countries | Landslide location accuracy varies from exact to 50 km range. The coarsest location accuracy used was 10 km. | 1988–2015, most data 2007–2015 | Daily for most data points |
| Climate Hazards Infrared Precipitation with Stations (CHIRPS) (Funk et al., 2015) | Station-corrected gridded precipitation data derived from cloud temperature observed using infrared satellite observations | 50°S to 50°N | 0.05° ($\sim$ 5.5 km) | 1981–2020 | Daily |
| MODIS Burned Area (Giglio et al., 2018) | Dates on which a pixel was burned, derived from NASA's MODIS Terra and Aqua satellites. The product uses a reprocessing algorithm that combines changes in burn-sensitive vegetation index and active fire locations. | global | 500 m | 2000–2020 | Daily |
| Daymet Precipitation and Snow Water Equivalent (Thornton et al., 2014) | An alternative precipitation dataset based on station data and topographic information | North America | 1 km | 1980–2020 | Daily |

First, the cluster tree was truncated at 30 clusters, after which all the clusters with fewer than 100 data points or less than $5\%$ burned sites were eliminated. Notably, two commonly studied regions for mass movements - Europe and Australia (e.g. Van Den Eeckhaut and Hervás, 2012; Nyman et al., 2011) – were eliminated at this stage due to a lack of verifiable post-wildfire mass movements available in the GLC. Cases where two nearby regions both had lower numbers of mass movements, for example, Central America and Caribbean/Venezuela, were joined manually. Finally, the largest region, encompassing Western US and Canada, was split into three sub-regions based on an additional identical clustering process over this sub-domain. The final regions are shown in Fig. 1 panel (a). The Pacific Northwest of North America was included even though the percentage of burned sites is lower than threshold, but at $4.4\%$ it was nearly double the highest percentage among the eliminated regions ($2.25\%$ in the Eastern US). Some mass movements were not included in any of the final regions. These events were not, however, eliminated from any analysis of all mass movements.

## 2.3 Fire data

For each mass movement, a circle centered at the mass movement location and with a radius of the location accuracy was computed and each $500\,\mathrm{m}$ pixel within the circle was extracted from the MODIS Burned Area dataset (Giglio et al., 2018). Fire affects the landscape over a large range of temporal scales in different settings. Previous studies suggest that the post-wildfire increase in mass movement susceptibility peaks within the first six months, but that a second time period of increased susceptibility can appear at 3 years or even longer as a result of root decay (DeGraff et al., 2015; Gartner et al., 2014). Landslides were classified as burned if any part of the area where the mass movement occurred was burned at some point within the three years prior to the event to capture both waves of increased susceptibility without over-identifying mass movements areas where fires occur every few years. The fraction of pixels that were burned over the 3-year antecedent period was then computed, and mass movements classified as burned if there was any overlap between burned areas and the mass movement circle. As a result of this analysis, 489 mass movements ($9.2\%$) were categorized as potential post-wildfire events.

While this method of identifying post-wildfire mass movements ensured that all post-wildfire mass movements were classified as burned, the low spatial accuracy of many of the mass movement locations leaves open the possibility that some mass movements occurred near a recent fire but not within the fire perimeter. Due to uncertainty in the exact location of many of the mass movement locations, both false positive and false negative errors in burn history classification are possible. Some mass movements classified as burned may have occurred near a recent fire but not within the fire perimeter, or conversely some mass movements classified as unburned may in fact have been located inside a fire perimeter but near the edge. However, by classifying mass movements as burned if any part of the potential location was burned limits the potential for false negative errors while increasing the possibility of false positive errors. For this reason we refer to mass movements as 'burned' instead of post-wildfire in this analysis. Also important to note is that false positive burned classification is a function of both the burned fraction and the conditional probability of mass movement occurrence given that a fire has occurred. False positives are therefore less likely for mass movements with better location accuracy, which made up a larger proportion of mass movements in the regions within the US and Canada than other regions. Fig. 1 shows the distributions of burned fractions for each region. Note that in Central America and Southeast Asia, very few sites have above 10 % burned fraction (see Fig. 1 panels (c) and (d)

inset plots). This could be due to those regions having lower mass movement location accuracy, resulting in a higher likelihood of false positive post-wildfire mass movements.

To explore the effects of variability in location accuracy and mass movement type within the GLC, validation analyses were performed to quantify the extent of errors due to these factors. Firstly, the percentages of burned sites in each region were computed for each location accuracy. Subsequently, the results of the Mann-Whitney hypothesis tests comparing pre-landslide precipitation percentiles were duplicated splitting the data in the high- and low-accuracy groups ($<= 1$ km and $> 1$ km respectively). The number of days with statistically significant differences in precipitation percentile in the 14 days prior to the mass movement and 7 days are computed in each group. Finally, a similar analysis compared debris flows (labeled as 'debris flow' or 'mudslide' in the GLC) and other types of mass movements.

## 2.4 Precipitation data

Time series of precipitation at the mass movement sites were obtained from the CHIRPS precipitation dataset (Funk et al., 2015). CHIRPS is a gauge-corrected global precipitation database derived from satellite-based cloud temperature measurements. The CHIRPS dataset was chosen because of its global coverage and relatively long climatological record (1981-present). Though the $\sim 5.5$ km resolution of CHIRPS may present challenges in capturing high-intensity storms that sometimes trigger landslides (Hong et al., 2007), Gupta et al. (2020) found that CHIRPS performed well in detecting extreme precipitation across India. Furthermore, this resolution matches the 5 km resolution of the plurality of records in the GLC. Precipitation was averaged for each mass movement location within the radius of the provided location accuracy. Additional pre-processing steps described below were performed to distinguish anomalously high precipitation events from potential seasonal shifts and climatic differences across sites.

Mass movements can be triggered by intense and short storms, by long storms of lower intensity, or somewhere in-between. A 7-day running average of antecedent precipitation was computed to enable direct comparison of the mass movements triggered by storms a range of durations. While including an estimate of the soil moisture was outside the scope of this study, 7-day antecedent rainfall indices consisting of a weighted average of precipitation over the 7-day time period have been used by other modelling studies as a surrogate for soil moisture in a combined indicator of landslide susceptibility (James and Roulet, 2009; Kirschbaum and Stanley, 2018). Furthermore, 7-day sums of precipitation have been found to perform better than other durations in threshold models of landslide occurrence (Krkač et al., 2017; Garcia-Urquia and Axelsson, 2015). Figure 1 panels (e) and (f) show these 7-day cumulative-precipitation percentiles, as well as the climatological seasonal average precipitation, revealing that the Western US is dominated by dry summers, while the lower-latitude regions exhibit wetter summers and in some cases monsoons.

Upon computing the CHIRPS precipitation measurements for each event, we noted that some of the categorized rainfall-triggered mass movements in fact had no recorded antecedent precipitation in the 7-day window. We screened these such mass movements from the analysis. Figure 2 shows a quality control sub-analysis for the California/Nevada area to investigate the need for data screening on the basis of inconsistencies between the reports of rainfall-triggered mass movements and the precipitation record. This region was chosen for the quality control analysis, because of its high density of precipitation data

and variety of climate conditions, useful for identifying erroneous mass movement precipitation. We found $14\%$ (73 of 533) of the mass movements in this region had no triggering precipitation event recorded in the CHIRPS data. Since the GLC contains only rainfall-triggered mass movements, the lack of precipitation in these cases was likely a result of errors in either the precipitation data or mass movement data.

A comparison with the Daymet precipitation dataset over the same domain revealed that the two precipitation datasets frequently did not agree on these zero-precipitation mass movement events, suggesting that the problem largely originated from the precipitation data themselves. Daymet is higher-resolution than CHIRPS (1 km vs. 5.5 km) and is based on precipitation gauge measurements. The extent of Daymet is limited to North America and thus is only used for validation in the California area. Furthermore, the concentration of data points on the x and y axes of Fig. 2 suggests that disagreements on precipitation occurrence are distinct from disagreements on the non-zero amounts of precipitation and potentially a separate source of error. To limit the effect of these inconsistent data points on the results, all mass movements worldwide with no measured precipitation in the six days before and one day after the event were removed from the global study (367 of 5680 or $6.5\%$ removed for a final $n = 5313$).

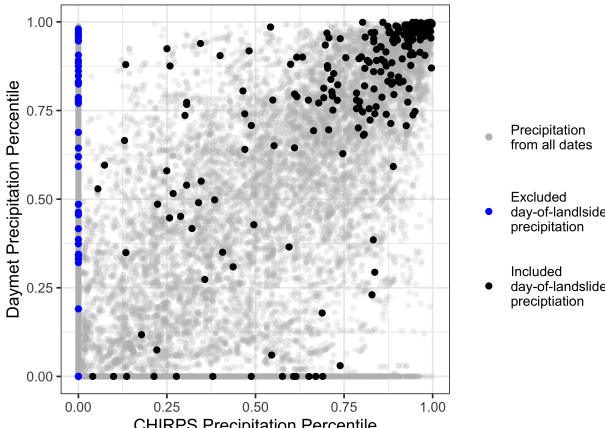

**Figure 2.** Seven-day precipitation percentiles for Daymet versus CHIRPS products computed for the six days before and one day following recorded California/Nevada mass movements. Blue and black points show the screened and included mass movements, respectively, whereas cumulative precipitation from the rest of the available record is shown in grey.

Precipitation data were further processed to facilitate the comparison of mass movement-triggering events across a variety of seasons and climates. In an initial analysis of the precipitation data, we were unable to distinguish between normal seasonal increases in precipitation and specific mass movement-triggering precipitation. In order to isolate triggering storms, it was necessary to normalize for both location and time of year. We accomplished this by computing a 30-day rolling percentile of the 7-day running precipitation values based on 38 years of historical precipitation climatology from 1981–2019 for each location. Percentiles have been used to compare landslide-triggering precipitation across larger, e.g. country-sized regions (Kirschbaum et al., 2020; Araújo et al., 2022) in order to control for differences in climate or precipitation data source. For

this study, the percentile produced a uniform distribution of precipitation ranging from 0 to 1, controlling for geographic and seasonal differences. As a result, anomalous precipitation events are highlighted, facilitating the comparison of mass movement triggers across locations and seasons.

## 2.5 Precipitation percentile experiment

This experiment compares the 7-day precipitation percentile in the burned and unburned groups in the time leading up to a mass movement. The percentile indicates the degree to which mass movement-triggering storms were exceptionally large and also serves as a proxy for relative mass movement susceptibility. A one-sided Mann–Whitney hypothesis test was used to ascertain whether the precipitation percentiles of burned sites were less than the precipitation percentiles of unburned sites. Deviations between the burned and unburned groups defined by a p–value less than $0.05$ on the Mann–Whitney test indicate statistically

significant differences in the mass movement susceptibility of the two groups. The null hypothesis of the Mann–Whitney test was that the distribution of precipitation percentile of the burned sites is generally greater than or equal to the distribution of precipitation percentiles of the unburned sites (Helsel et al., 2020). Percentiles are by definition uniformly distributed rather than normally distributed, making the Mann–Whitney test, since it does not require normal distribution, the most appropriate hypothesis test for these data. However, since zero-precipitation periods are excluded, this method cannot account for dif-

ferences in the frequency of precipitation across different climates, but rather reflects differences in the magnitude of 7-day precipitation totals.

## 2.6 Bootstrapped samples experiment

In order to evaluate how anomalous the precipitation events preceding burned and unburned landslides were to "typical" local climate conditions at the mass movement locations, we compared them to bootstrapped samples from other years to obtain

a clearer signal. One hundred samples were taken from the 38-year precipitation records to match the locations and DOY of the observed mass movements, but from randomly selected years ($n$ =the smallest number above 100 that ensured each site was included in the same number of samples). Sampling was repeated for burned and unburned groups within each region as well as for all the mass movements in the study. These samples are representative of precipitation for a particular number of days before the mass movement and serve as a control dataset with which to compare the pre-landslide precipitation. Next,

the observed event-year precipitation across all sites in the group was tested against each bootstrap sample using a Mann–Whitney test, with the null hypothesis that the sample median precipitation percentile was less than or equal to the median of the precipitation percentiles from that day of the year in the entire record from 1981–2020. This produced a distribution of p–values that represent the likelihood that the precipitation leading up to the mass movements varied from the control baseline.

This sampling method, though more complex, helps to reduce noise in the hypothesis test results due to different sample sizes

in different regions. It also provides more information on general mass movement susceptibility of each region rather than only the relative susceptibility of burned and unburned sites. Finally, it includes measurements of zero precipitation, which were eliminated from the direct comparison because of long-term climatic differences in precipitation frequency between burned and unburned sites in all regions.

## 2.7 Mass movement seasonality experiment

The probability of landslide occurrence in a given temporospatial domain varies throughout the year (Stanley et al., 2020); we refer to this annual pattern for a given domain as mass movement seasonality. We hypothesize that wildfire alters mass movement seasonality. To test this hypothesis, we estimated precipitation frequency at the mass movement sites over time by computing the fraction of sites in the burned and unburned groups that had precipitation on any given day. As with the percentiles and the bootstrap p–values, frequency estimates were computed relative to the mass movement event rather than by

calendar date, resulting in time coordinates measured in 'years before the event'. Precipitation frequency was estimated for two years before and after the mass movement in order to highlight changes in the magnitude and phase of the precipitation pattern. We found that in most regions there was a long-term difference in the mean annual precipitation frequency, likely because fires occur more often in areas with drier climates (Liu et al., 2014) and drought (Balling et al., 1992; Gudmundsson et al., 2014). These persistent differences between burned and unburned sites were removed by standardizing the mean precipitation

frequency for both the burned and unburned groups, that is to say subtracting the mean and frequency and dividing by the standard deviation. Finally, we took a 90-day running average to reduce noise in the data and thereby make it easier to visually identify any long-term shifts in mass movement occurrence. These frequency estimates are not normalized by season, which means that unlike the previous two metrics they can be used to compare the degree of shift in the seasonality of mass movements at burned versus unburned sites relative to annual precipitation cycles.

Additional seasonality analysis was performed to provide insight into the times of year that mass movements occur at burned versus unburned sites. Kernel density estimates of mass movement occurrence throughout the year were compared between the burned and unburned groups. This seasonality analysis would highlight a shift from Fall to Spring but, in contrast with the frequency analysis, it does not indicate the precipitation conditions under which mass movements typically occur. Together, the frequency and seasonality analyses can show both the seasonal shift as well as any changes in mass movement occurrence

relative to annual precipitation patterns.

## 3 Results

### 3.1 Precipitation percentile experiment

The distributions of precipitation event percentiles for all the included mass movements are shown in Fig. 3. The precipitation percentile increases for all groups as the date of the landslide approaches, confirming that these rainfall-triggered landslides

are generally preceded by an increase in total precipitation depth. Notably, when considering all mass movements together (Fig. 3) the precipitation events that triggered landslides at burned sites were significantly smaller than those that triggered mass movements at unburned locations (Mann–Whitney test, $95\%$ confidence). At first glance, this difference supports the overarching hypothesis that wildfire does in fact increase mass movement susceptibility, since mass movements in the period after a fire can be triggered by less precipitation than might normally be required to cause mass movement. However, an

examination of each region separately reveals that the difference in precipitation percentile between burned and unburned

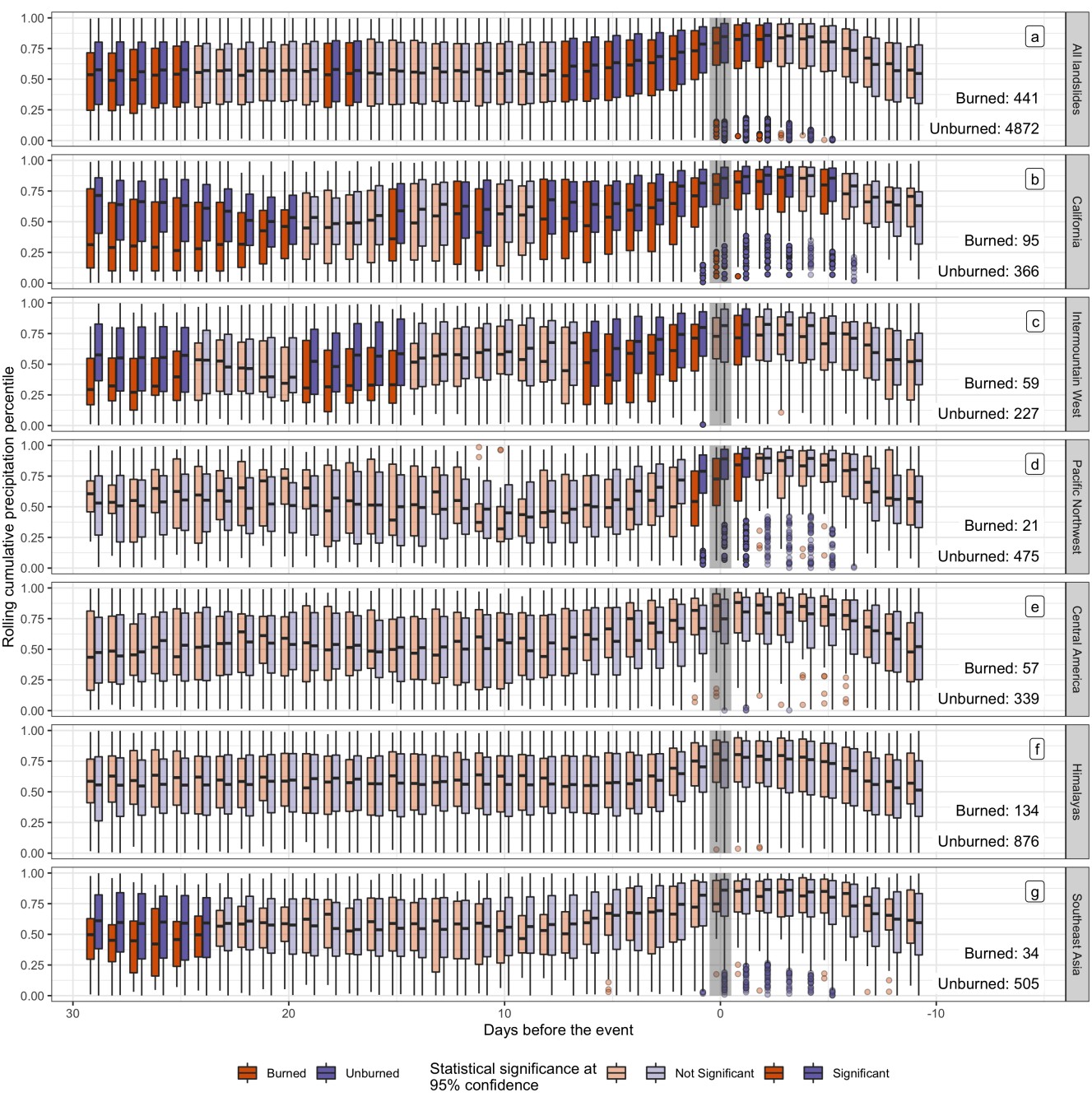

**Figure 3.** Seven-day precipitation percentile in the lead-up to mass movements for all mass movements in (a) and for the six individual regions labeled (b)–(g), whether classified as part of one of the regions or not. The day of the mass movement is indicated with a vertical grey column. Days where a significant difference was found between the burned and unburned groups are indicated in darker colors (Mann–Whitney hypothesis test, $p > 0.05$).

sites is present in some regions but not in others (see Fig. 3). The California area (Fig. 3 panel (b)) has a particularly strong signal, whereas tropical regions do not show any significant decrease between precipitation at burned and unburned sites or display the reverse effect of higher precipitation percentiles for unburned locations than burned locations. In total, these initial results suggest that post-wildfire landslides are isolated to areas, such as California, where such cascading hazards have been repeatedly observed.

Figure 4 shows p-values for Mann-Whitney hypothesis tests comparing precipitation percentiles for burned and unburned groups for high and low location accuracy groups of mass movements. High accuracy indicates less than 1 km. Several regions, such as California (Fig. 4 panel (b)) show substantial differences between the high-accuracy and low-accuracy p-values. Sample sizes of burned locations among the exact locations are low, ranging from 2 to 34 in each region, with overall only 3.7% of high-accuracy mass movements classified as burned (below the threshold used to exclude regions from this study). The low percentage of burned sites may partially account for high p-values among the high-accuracy group. An additional important consideration is the likelihood of a greater number of false positive burned sites among the low-accuracy group. Notably, the percentage of identified burned sites using this method increases with the location accuracy radius – globally 12.5% of low-accuracy mass movements were identified as burned in contrast with only 3.7% of high-accuracy mass movements.

Figure 5 shows the p-values of Mann-Whitney hypothesis tests, similarly to those performed for Fig. 3. The results in Fig. 5 are split into categories by mass movement type, with 'debris flow' and 'mudslide' types labeled as debris flows and all other types labeled as other. With the exception of the Pacific Northwest (Fig. 5 panel (d)), the mass movement type has limited impact on the number of days with significant differences ($p < 0.05$) in precipitation in the 14 days prior to the mass movement in regions with any such significant differences. For example, in California (Fig. 5 panel (b)), nine days have a statistically significant difference for both groups. In the Intermountain West eight days have a statistically significant difference for debris flows while similarly six days have a statistically significant difference for other types of mass movements.

### 3.2 Comparison of bootstrapped samples and pre-landslide precipitation

Figure 6 highlights the increase in precipitation in the days before a mass movement relative to historical amounts for that location and time of year, i.e., relative to climatology, offering a robust assessment of the mass movement precipitation departure. The Mann–Whitney p–values comparing the precipitation record on each day to each of the $\sim 100$ samples are shown in 6 panels (a)–(g). Mass movement events have been split into burned and unburned groups (shown in orange and purple respectively) for six regions and for all mass movements in the study. Bootstrapped samples were drawn from the same DOY and locations as the mass movements but from a randomly selected year. In panels (a)-(g), box plots of p–values represent the degree to which the mass movement-triggering precipitation differed from climatological precipitation with lower p-values indicating a more significant difference between the two precipitation distributions. The Mann-Whitney tests were directional, so only differences where the precipitation is greater than would be expected result in low p-values. Examples of the kernel density estimates of each bootstrap sample as compared to the precipitation on the day of the mass movement are shown in Fig. 6 panels (h)–(u) to better illustrate the comparisons made by the hypothesis tests in panels (a)–(g). Each orange or purple curve was tested against the black curve to obtain the boxplots of p–values at 0 days before the mass movement. A clear difference

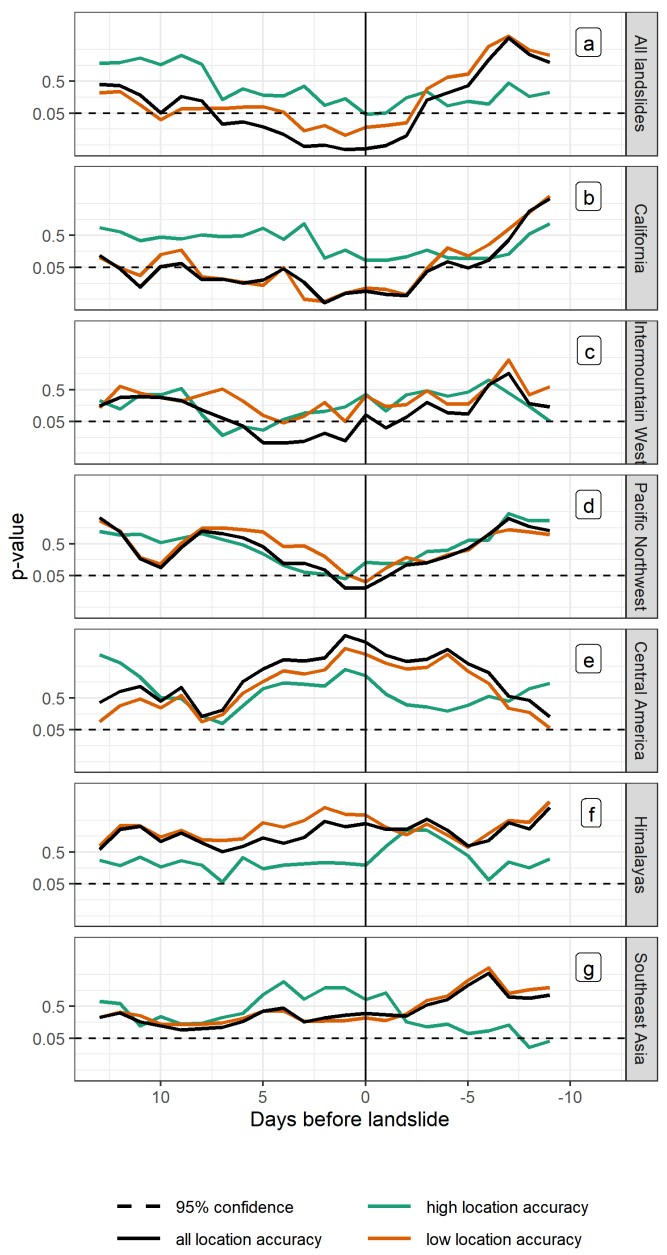

**Figure 4.** p-values for Mann-Whitney hypothesis tests comparing precipitation percentiles at burned and unburned sites. The thick black line shows the p-values for all mass movements, while green and orange lines show high (1 km or less) and low (greater than 1 km) location accuracies. A horizontal black line shows the p=0.05 significance threshold, while a vertical black line indicates the day of the event.

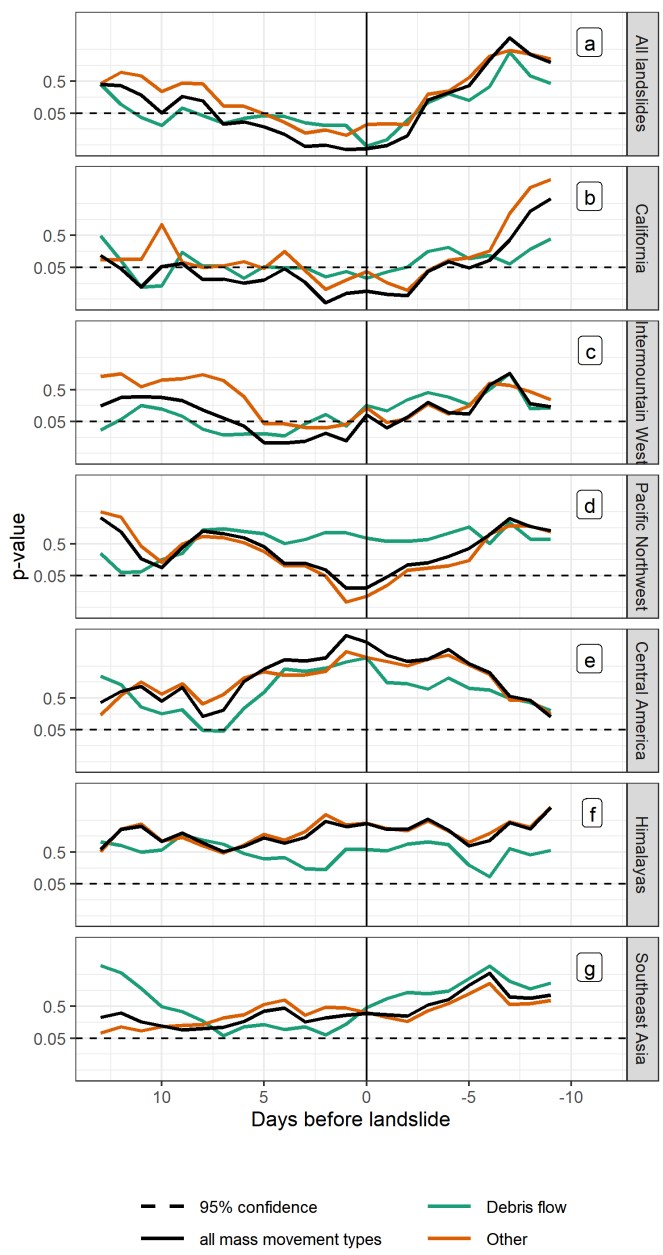

**Figure 5.** p-values of Mann-Whitney tests comparing mass movement-triggering precipitation percentiles at burned and unburned sites. The black line shows results for all mass movements, while debris flows and other mass movements are shown in green and orange respectively. A horizontal black line shows a 95% confidence level for the hypothesis test, and a vertical black line indicates the day of the mass movements

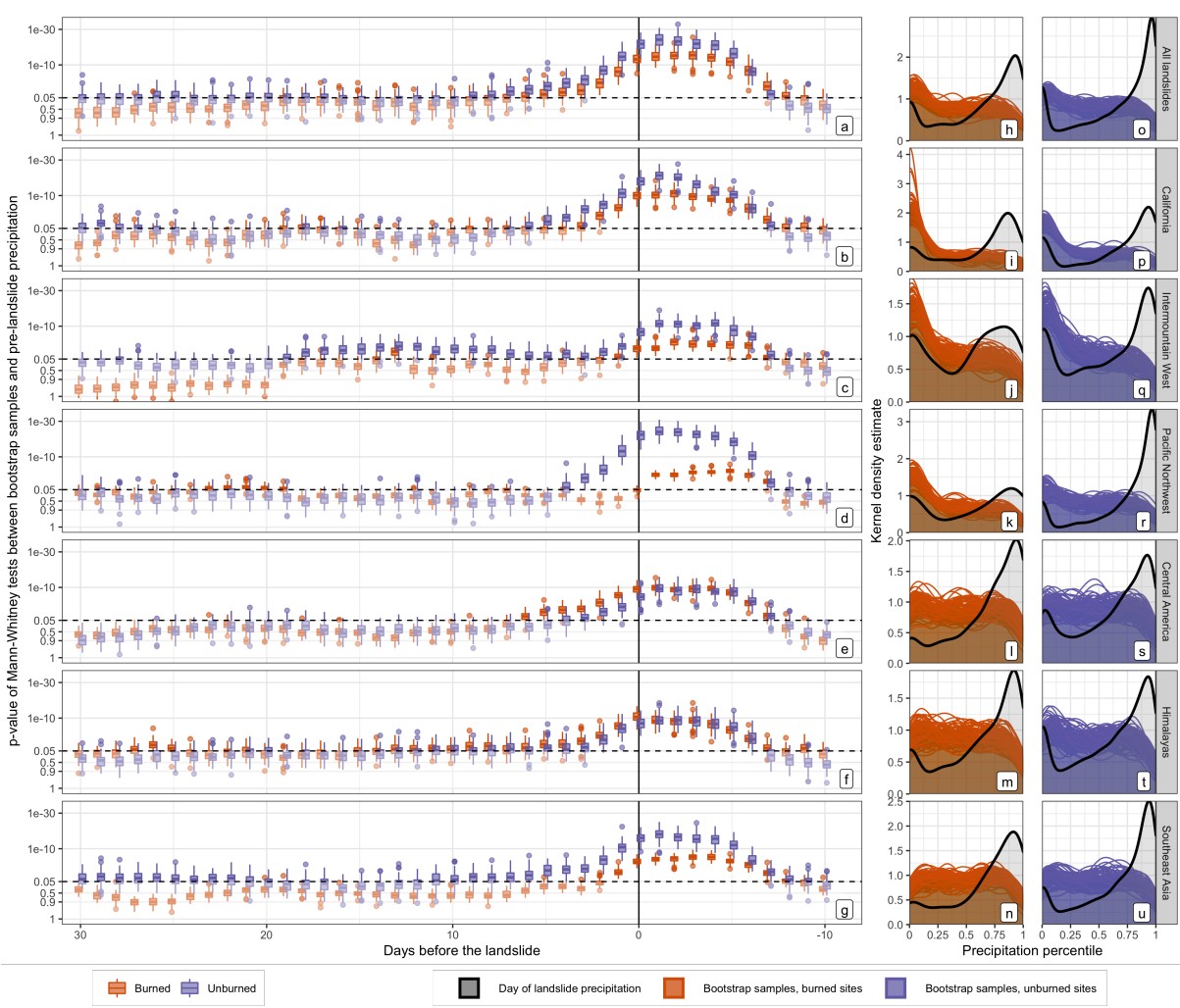

**Figure 6.** p-values of Mann–Whitney hypothesis tests comparing mass movement-triggering precipitation relative to 100 bootstrapped samples (n 100 for each sample) drawn from a 38-year precipitation record from the mass movement locations. The y-axes are shown with a probit transform to expand the section of the axis where p-values are below 0.05 (significant at 95% confidence, shown as a dashed black line). The y-axis has also been inverted so that larger differences in precipitation (lower p-values) are higher on the y-axis for consistency with the percentile plots in Fig. 3. In panels (h)-(u), an example of the kernel density estimate (kde) for day-of-landslide precipitation in black separated by burned and unburned groups is compared with kdes of all bootstrapped samples in orange (burned group) or purple (unburned group).

between burned and unburned sites is shown for the same regions as in Fig. 3, but with the addition of Southeast Asia. Beyond the emergence of a signal in Southeast Asia, additional differences between regions in the timing of precipitation in the period leading up to the mass movement are visible in Fig. 6 panels (a)–(g) that were not clear in the precipitation percentile analysis.

Different storm timing is apparent among the regions, and between the burned and unburned sites of the same region. Firstly, in California and Southeast Asia (Fig. 6 panels (b) and (g)) )), we see a similar pattern where relative precipitation at unburned sites is consistently higher than at burned sites. Nonetheless, for both burned and unburned sites, the rise in precipitation takes place over a similar amount of time (approximately 5 days). Curiously, unlike in California, the bootstrap analysis reveals a long-term difference between burned sites and unburned sites in the Mann–Whitney p–value for Southeast Asia despite location-specific normalization, suggesting that the mass movements at unburned locations might be primarily triggered in years that are wetter than usual on a monthly or seasonal scale. In the Pacific Northwest (Fig. 6 panel (d)), the precipitation at the burned sites does not become significantly larger than climatology until the day of the mass movement. The Mann–Whitney p–values for the burned group remain well above 0.05 just days before the mass movement as the p–value for the unburned group begins to fall. Under the assumption that shorter storms are associated with runoff-driven mass movements while longer storms that allow more time for the soil column to saturate are associated with infiltration-driven mass movements, this difference in storm timing could reflect that in the Pacific Northwest the burned mass movement locations are largely runoff-driven while mass movements at unburned locations are infiltration-driven (Cannon and Gartner, 2005). The Mann–Whitney p–values for the burned group remain well above 0.05 just days before the mass movement as the p–value for the unburned group begins to fall. In the Intermountain West (Fig. 6 panel (c)) antecedent precipitation for the burned group is generally characterized by a dry spell going back thirty days or more. In this region, thirty to twenty days before the mass movement p–values for burned sites are consistently above 0.9, suggesting a high likelihood ($> 90\%$) that there was less precipitation than usual during that time. During the same period, the p-values at unburned sites remain close to 0.05

### 3.3 Landslide and fire seasonality experiment

Figure 7 shows the seasonality of fires and mass movements at each site, in addition to the length of time elapsing between the fire and the mass movement. Landslides in several regions, especially California and the Himalayas, tend to occur at the same time of year. This time of year, for the regions where it exists, will be referred to as 'landslide season.' Similarly, nearly all of the regions have a fire season, which is most clearly visible in the black rug at the top of each panel in Fig. 7. Figure 7 panel (a) shows that fires occur nearly year-round when considering all regions together, but the other panels in Fig. 7 show that within any particular region, fires occur only during a distinct time of year. However, the delay between fire and mass movement is not consistently equal to the length of time between fire season and the following mass movement season. The mass movements are distributed such that $53\%$ occur within a one year after the fire. Since both mass movements and fires have seasonal patterns, the typical delay between fire and mass movement for each region appears to be primarily related to the relationship between fire season and mass movement season. For example, California has a long fire season and a shorter landslide season, and so when fires occur at the end of winter, immediately after mass movement season, there is typically a longer delay before the mass movement than when fires occur immediately before mass movement season. By contrast, in the

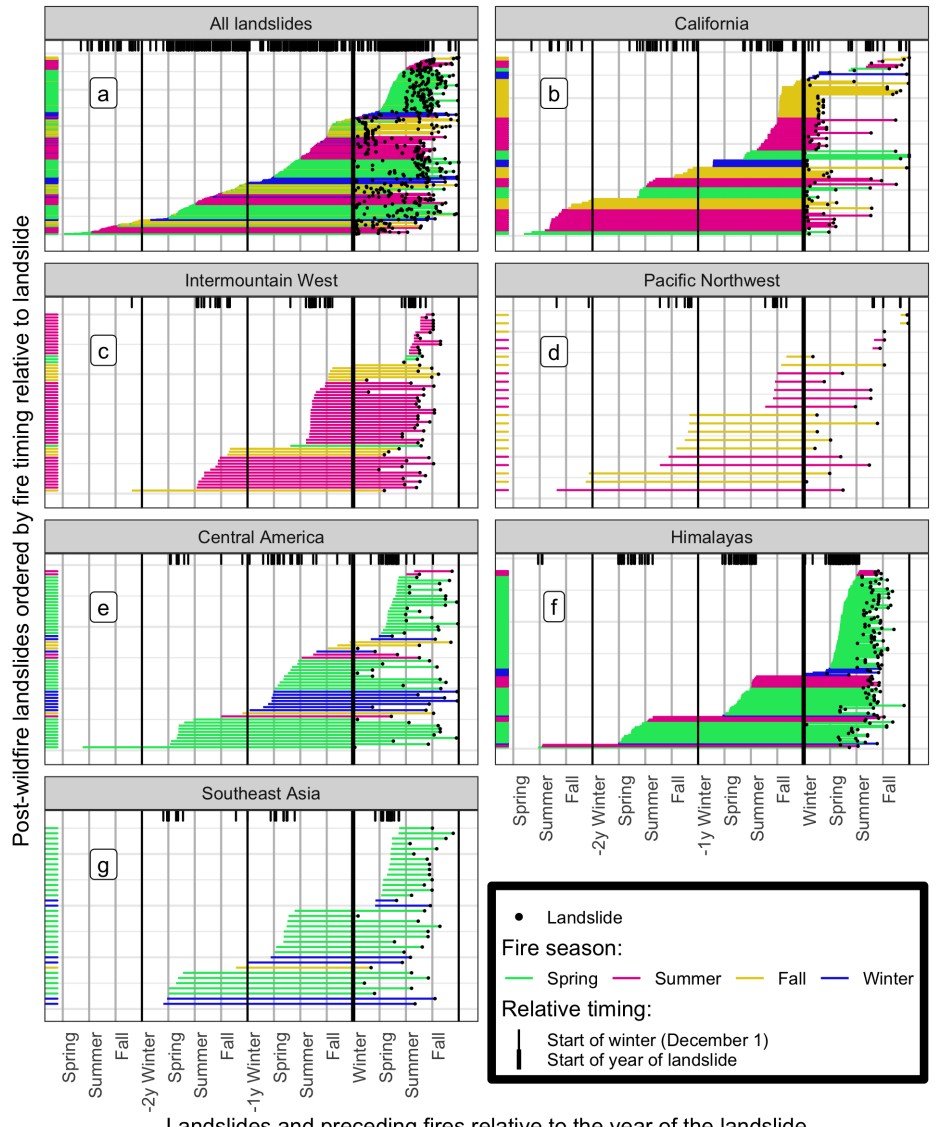

**Figure 7.** DOY of mass movements, DOY of fires, and the length of time in between fire and mass movement by region. Each horizontal line represents one event, arranged on the y-axis in order of the length of the delay between wildfire and mass movement. Black dots on the right show the day of the year the mass movement occurred, and horizontal lines represent the duration of time elapsed in between the fire and the mass movement. Lines are colored by the season of the fire and are ordered by the day of the fire relative to the mass movement. The black lines, or rug, at the top of each panel as well as the colored rug on the left duplicate the day-of-year of the fires to highlight seasonal patterns.

Himalayas the delay between fire and landslide is relatively uniform due to a shorter fire season that does not overlap with
the mass movement season. In general, the mass movements occur during the period of greatest rainfall, such as the winter
in California and the summer in the Himalayas (see Figure 1 for regional precipitation climatology). The seasonal pattern
of post-wildfire mass movements is to some degree determined by an interaction between fire seasonality and precipitation
seasonality.

Figure 8 shows the p-values of Mann-Whitney tests comparing precipitation percentiles of groups of mass movements with
different timing relative to wildfire with precipitation percentiles of mass movements at unburned sites. Landslides at burned
sites were divided into two groups: within one year after a wildfire, mass movement between one and three years after a
wildfire. In California and the Pacific Northwest of the US (Fig. 8 panels (b) and (d)), the p-values are similar among the two
timing groups. By contrast, in the Intermountain West of the US (Fig. 8 panel (c)), the lower precipitation percentiles at burned
sites are only statistically significant at the time of the for mass movements occurring 1-3 years after a wildfire. However,
precipitation is significantly lower in the 'less than one year' group in the seven-to-three days before the mass movement. In
Central America, the Himalayas, and Southeast Asia (Fig. 8 panels (e), (f), and (g)), differences between burned and unburned
sites are not statistically significant for either group.

Figure 9 shows differences in seasonality between burned and unburned mass movement seasonality on the right and the
results of the precipitation frequency analysis on the left. The kernel density estimates on the right show changes in the seasons
(e.g. Fall or Winter) in which landslides at burned and unburned sites occurred. By contrast, the analysis on the left shows when
landslides in each group tended to occur relative to the times of year with greater precipitation frequency. While all regions
except for Central America (Fig. 9 panel (l)) display some kind of shift in seasonality between burned and unburned mass
movements in right-hand panels of Fig. 9 ((h)–(n)), the magnitudes and directions of these shifts varies by region. Interestingly,
the regions with clear shifts in seasonality have shifts of different directions, i.e. earlier or later in the year, and magnitudes,
i.e. a few weeks to half a year. In the Southeast Asia (Fig. 9 panel (n)), mass movements at burned sites happen in the summer
rather than the winter for unburned sites, a 6-month shift. In contrast, mass movements in the Intermountain West (Fig. 9 panel
(j)), burned mass movements appear to happen in the spring while unburned mass movements occur in the winter, a 3-month
shift later in the year. In California (Fig. 9 panel (i)), by contrast, burned landslides are shifted earlier in the year and by only
a few weeks, with both burned and unburned mass movements occurring primarily in the fall and early winter. Finally, in the
Pacific Northwest (Fig. 9 panel (k)), it appears that some of the burned mass movements occur in the usual mass movement
season of fall and early winter, while another peak lies 6 months away at the beginning of summer.

The precipitation frequency in Fig. 9 panels (a)–(g) highlights differences in when mass movements tend to occur relative to
the wetter parts of the annual precipitation cycle between burned and unburned groups. A curve for burned sites that is shifted
slightly to the right of the corresponding curve for unburned sites, as is the case for the burned group precipitation frequency
in the California region (Fig. 9 panel (b)), indicates that burned landslides occurred earlier in the rainy season. In California
(Fig. 9 panel (b)) burned mass movements are clearly shifted to a period earlier in the year with more frequent precipitation,
i.e. earlier in the wet season, although the shift is larger in California. This provides evidence confirming our hypothesis that
wildfire increases mass movement susceptibility in these regions, since it suggests that a smaller precipitation trigger that might

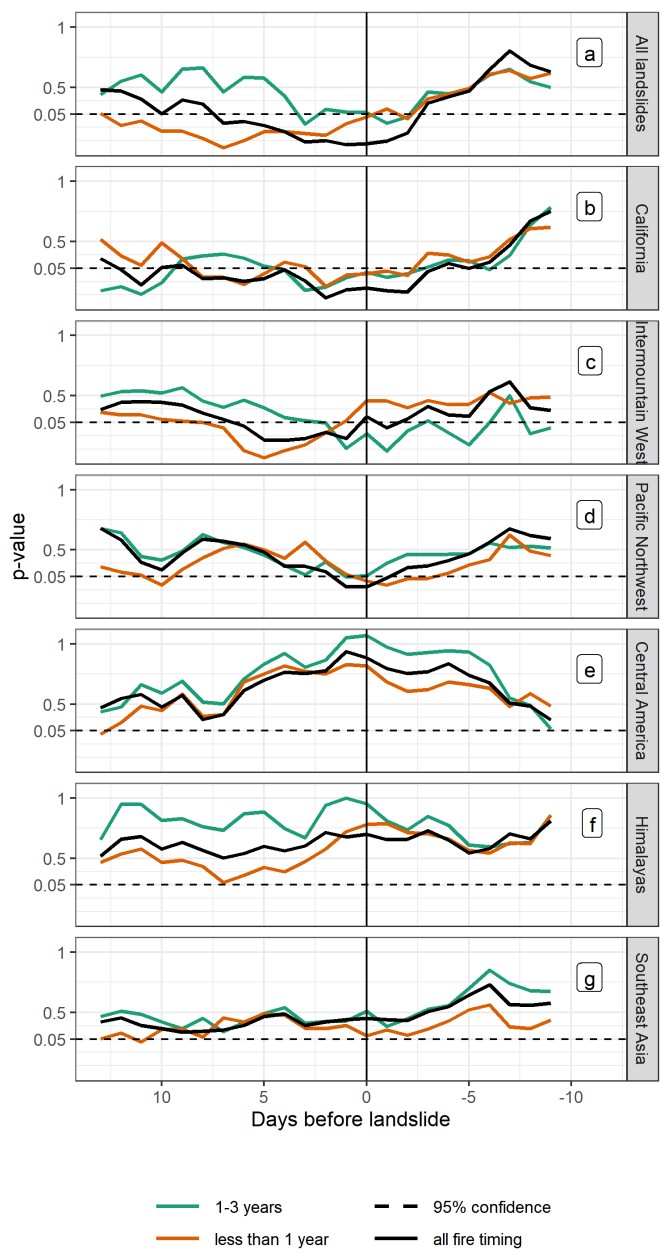

**Figure 8.** p-values for Mann-Whitney hypothesis tests comparing precipitation percentiles at burned and unburned sites. The thick black line shows the p-values for all mass movements, while orange and green lines show mass movements occurring within one year of a wildfire and between one and three year of a wildfire respectively. A horizontal black line shows the $p = 0.05$ significance threshold, while a vertical black line indicates the day of the mass movement.

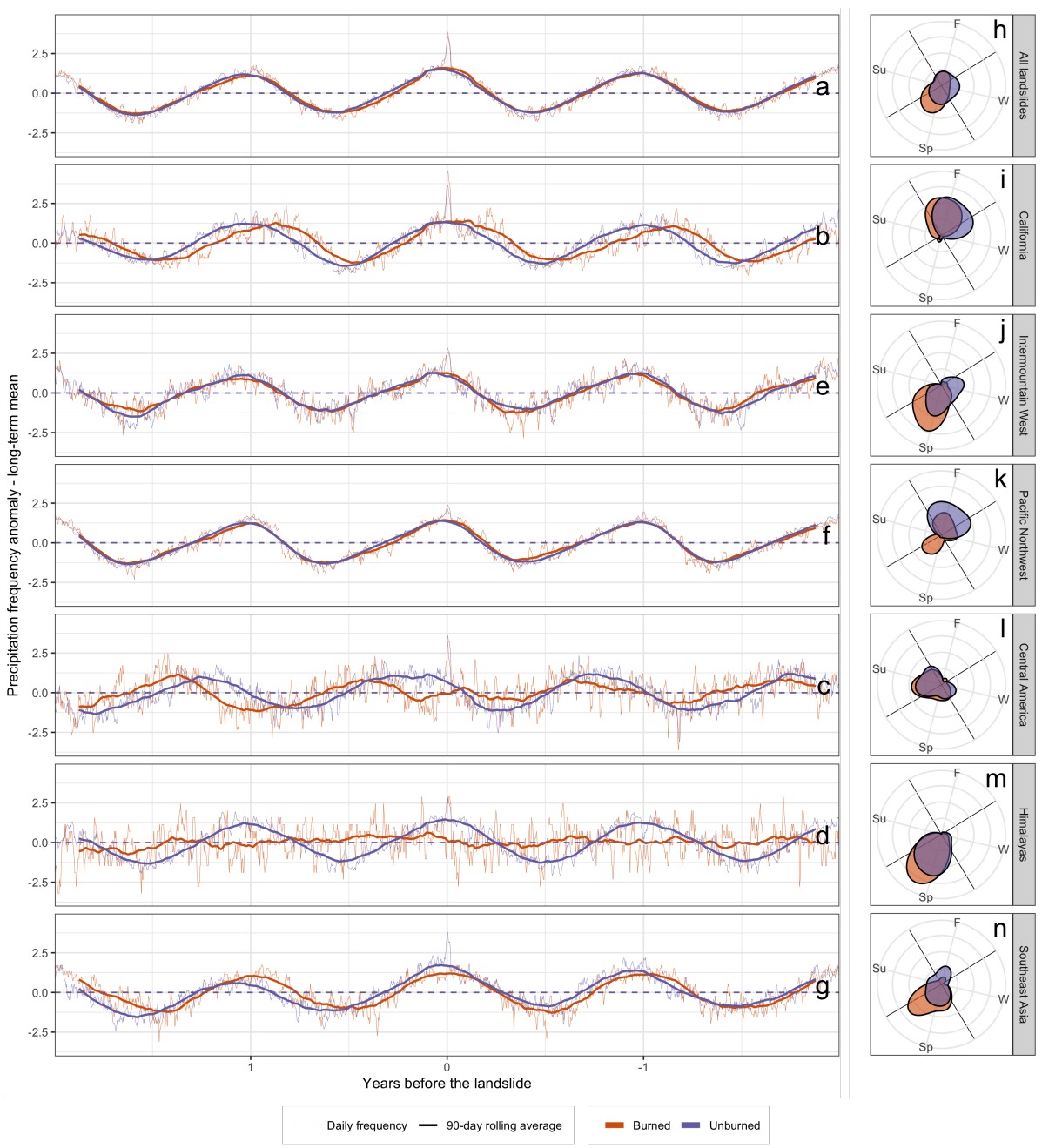

**Figure 9.** Precipitation frequency anomaly relative to the long-term mean aligned by the mass movement date. In panels (a)(g), frequency is shown both daily and smoothed with a 90-day moving average to highlight shifts. Daily precipitation frequency is represented as thin lines in orange and purple (burned and unburned groups) while the 90-day average is a thicker line. The long-term mean has been removed from all the frequency curves. Landslides are in burned and unburned groups for each region separately and for all mass movements. In panels (h)–(n), the kernel density estimate of mass movements by the time of year is shown for both the burned and unburned groups in a radial plot.

be found earlier in a wetter part of the year is required to trigger a mass movement after a fire. The Intermountain West (Fig.

9, panel (c)) also has a pronounced seasonal shift, but in this case the shift is much larger, so much so that the burned mass movements in this region appear to occur as a result of a large storm in the middle of a dry part of the year. Other regions (Pacific Northwest in panel (d), Southeast Asia in panel (g), and Central America in panel (e)) show differences in the magnitude of the annual cycle in precipitation frequency, but no shift in seasonality. These magnitude changes are not consistent in direction or degree across regions. In Southeast Asia, where Fig. 9 panel (n) shows a shift in seasonality but panel (g) does not show a

shift relative to the wetter parts of the year, these results suggest that there could be a spatial or climatic bias to the locations of burned landslides that is causing the seasonal difference.

## 4  Discussion

The results of this study suggest that while post-wildfire mass movements are associated with shifts in the magnitude, timing, and seasonality of storms relative to other mass movements, these effects are not consistent across regions. Globally, there are

clear differences in the percentiles of mass movement-triggering storms (see Fig. 3), with mass movements in burned areas often triggered by comparatively smaller storms. At first glance, this supports the hypothesis that fires increase mass movement susceptibility, since a smaller precipitation trigger is sufficient to cause a mass movement. However, this trend is largely driven by the California region and to a lesser extent the Intermountain West and Pacific Northwest of North America. In Central America/Caribbean, Southeast Asia, and the Himalayan regions there is an increase in rainfall relative to climatology leading

up to the mass movement, but there is no significant difference between relative precipitation depths based on fire history. The original percentile analysis includes only wet days; the bootstrap analysis takes into account both wet and dry days.

Differences in the mass movement-triggering storms relative to their precipitation climatology shown by the bootstrap analysis (Fig. 6) confirm the results of the original percentile analysis, with two notable exceptions. In Southeast Asia, the bootstrap analysis indicates that the burned sites had smaller precipitation triggers relative to climatology despite no significant difference

in the first analysis between the wet-day precipitation percentiles. This discrepancy suggests that there may be a precipitation frequency bias between burned and unburned sites in this region. In addition, burned locations in the Pacific Northwest appear to be associated with rainfall that began closer to when the landslide occurred (Fig. 6 panel (d)). This raises the possibility that mass movements at burned sites in this region are caused more often than in unburned locations by runoff instead of infiltration. More information is needed, for example about the antecedent soil moisture at these locations. This result is consistent with

previous research suggesting that post-wildfire debris flows are predominantly triggered by runoff-driven erosion as a result of shorter and more intense storms in the Western US ($76\%$ Cannon and Gartner, 2005); however in that case we would have expected to see a similar pattern in the California and Intermountain West regions.

Many of the mass movements at burned locations in the Intermountain West (Fig. 6 panel (c)) appear to be particularly susceptible to shorter-duration storms that occur after a dry spell stretching from thirty to twenty days before the mass movement

and possibly even further back in time. A similar pattern of low frequency precipitation followed by a sharp spike can be seen in the burned locations in Fig. 9 panel (c). One possible explanation is that dry, recently burned soil is particularly erosive in

those areas. An example of drought conditions contributing to a landslide is described by (Handwerger et al., 2019). These differences are also due in part to the different regional climates, with the California and Pacific Northwest regions having more clearly defined longer-duration rainy seasons, relative to the more variable and sporadic precipitation seasonality of the Intermountain West.

Different combinations of fire season, mass movement season, and any overlap between the two may be an important driving factor in the degree to which fires increase mass movement susceptibility. For example, in places where the wet season begins towards the end or immediately after fire season, such as the Intermountain West, California, and the Himalayas, the landscape has no time to recover from the fire before mass movement season begins and therefore burned locations may be much more susceptible (see Fig. 7 panels (b), (c), and (f)). On the other hand, in regions like the Pacific Northwest, Central America, and Southeast Asia (Fig. 7 panels (d), (e), and (g)), where mass movement season is not as well defined, it is more likely that the landscape could at least partially recover before a triggering storm occurs.

Some of the regions that did not display a significant difference in percentile nonetheless showed a shift in the timing of burned mass movements relative to their respective annual pattern of precipitation (see Fig. 9 panels (h)–(n)). The various types of shifts in landslide seasonality are likely reflective of the different effect of fires. A shift of the mass movement season to slightly earlier in the year, such as was noticeable in California and the Himalayas (see Fig. 9 panels (i) and (m)) supports the hypothesis that wildfire increases mass movement susceptibility because it suggests that fewer or smaller precipitation events earlier in the season are sufficient to trigger a mass movement. The Intermountain West (Fig. 9 panel (j)) also has a pronounced seasonal shift, but in this case the shift is much larger and in the opposite direction: burned mass movements appear to occur an entire season later than unburned mass movement, falling in the driest part of the year instead of the wettest. This corresponds to the evidence from the bootstrap analysis suggesting that dried out soil or slow vegetation regrowth may be an important part of the post-wildfire mass movement mechanism in this region. Vegetation regrowth as a main control of mass movement susceptibility is supported by a study of mass movement occurrence in the San Gabriel mountains of the US by Rengers et al. (2020), in which the authors found that hillslopes with slower vegetation regrowth were more likely to have mass movements.

A similar trend to the Intermountain West in terms of seasonal shift is visible for some, but not all, of the mass movements in the Pacific Northwest (Fig. 9 panel (k)), suggesting perhaps that some of mass movements in that region would have been better categorized as part of the Intermountain West region. In Southeast Asia (Fig. 9 panel (n)) there also appears to be a seasonal shift similar to that of the Intermountain West, but it is not matched by a shift relative to the annual precipitation frequency pattern (Fig. 9 panel (g)). This suggests that the seasonality "shift" in Southeast Asia due to spatial bias in fire occurrence. Further study of variation in climate across this region is needed. Finally, Central America (Fig. 9 panel (l) has very similar precipitation frequency in burned and unburned locations. Since there is little difference between the precipitation frequency or magnitude (see Figs. 3 panel (e), 9 panel (e)) in this area, it is possible that there are many misidentified false positive post-wildfire mass movements in Central America, perhaps due to the generally low location accuracy in that region. It is also possible wildfire does not have as much of an effect on mass movement susceptibility in that region.

The timing of mass movements relative to wildfire may also influence the magnitude of triggering storms. While in some regions, such as California and the Pacific Northwest, timing does not have a major impact on precipitation percentile differ-

ences, the Intermountain West of the US displays two distinct behaviors depending on the timing of mass movements relative to wildfire. In the year immediately after a fire, the precipitation percentile is lower than for mass movements at unburned locations in the seven-to-three days before the mass movement, before rising to match precipitation percentile at unburned lo-

cations (see Figure 8 panel (c)). This pattern matches the result from Figure 9 panel (c) in which post-wildfire mass movements in this region appear to manifest as a large storm preceded by a period of infrequent precipitation. In contrast, timing appears to make little difference to the precipitation percentile in other regions.

Low mass movement location accuracy and lower number of burned mass movements may have also contributed to the lack of conclusive results in the Pacific Northwest, Southeast Asia and Central America. The regions outside the US and Canada

tended to have less accurate mass movement locations, and less accurate locations were also more likely to be marked as burned. Furthermore, less accurate locations were also more likely to be marked as burned, with a threefold increase in the percentage of mass movements identified as burned between high- and low-accuracy groups. This is because larger mass movement radii were more likely to contain burned area by chance alone, and hence become false positive post-wildfire mass movements, i.e. landslides that occurred nearby but not coincident to a burned area. This idea is supported by the lower cumulative burned

fractions within the regions outside the US and Canada (see Fig. 1 panels (c) and (d)). Though mass movement accuracy in the GLC is an approximate measure, introducing the possibility of false negative unburned sites, false positive post-wildfire mass movements nonetheless represent a major potential source of error in this analysis. These uncertainties introduce the possibility that some of differences in triggering precipitation percentiles between burned and unburned sites may be related to unique qualities of fire-prone areas rather than fire itself. Future studies using visible and other satellite imagery to pinpoint mass

movement locations and dates could help clarify the post-wildfire posterior landslide probability by essentially eliminating the location error. Furthermore, there is a body of research that uses GIS data such as slope or underlying lithography in combination with a statistical model like a classification tree or logistic regression to assess mass movement hazards (e.g. Felicísimo et al., 2013; Lee, 2007; Ohlmacher and Davis, 2003), including some focused on post-wildfire mass movements (Cannon et al., 2010). The introduction of such control datasets of confirmed unburned landslide locations would also allow

the use of additional variables like slope, land use, and aridity index to be incorporated into a model as part of an assessment of which properties of sites have the greatest influence on changes in mass movement susceptibility at burned sites.

## 5  Conclusions

Clear differences were shown between rainfall-triggered mass movements at unburned and unburned locations in the magnitude of precipitation triggers, the seasonality of mass movements, and the timing of triggering storms. These findings suggest that

wildfires increase susceptibility to mass movements, especially in regions of the Western US. However, they also suggest that post-wildfire mass movements are not a spatially uniform phenomenon. Both the mechanisms by which burned mass movements are triggered and the degree to which wildfire increases susceptibility varies by region.

The precipitation percentile immediately before a mass movement was found to be smaller at burned locations for all regions combined, as well as for the California, Intermountain West, and Pacific Northwest regions, but not for the others. This

result suggests greater mass movement susceptibility in those three regions following a wildfire. In California and the Pacific Northwest, mass movement-triggering storms tended to be shorter at burned locations, suggesting that these mass movements are more often runoff-driven than mass movements at unburned locations. In contrast, in the Intermountain West burned mass movement locations appear to be characterized by a dry spell of at least 20 days followed by a sharp uptick in precipitation, suggesting that burned and dry soil may be the most vulnerable to extreme erosion in that region. Finally, shifts in landslide

seasonality were noted in every region except Central America, although the characteristics of these shifts were not consistent among regions. In some regions such as California and the Himalayas, landslides at burned locations occurred earlier in the wet season, suggesting greater susceptibility to mass movements caused by fire. In other regions such as the Intermountain West and Southeast Asia, mass movement seasonality was shifted by 3 or 6 months, suggesting that the conditions resulting in mass movements differ in more fundamental ways at burned sites. For example, in the Intermountain West we posit that a

portion of post-wildfire mass movements may be caused by isolated intense thunderstorms on dry soil producing the observed pattern of mass movement-triggering storms in burned locations preceded by at least several weeks with limited precipitation. Among the unburned sites, by contrast, a pattern of mass movements occurring during the wettest part of the year suggests that saturation of the soil is a more important precursor.

Developing a better understanding of the ways in which mass movement hazards vary around the world is important for

mitigation efforts as well as predicting how mass movement hazards will respond to a changing climate. Data acquisition is a major barrier to this type of global analysis of mass movement statistics. Both precipitation and burn status are major sources of uncertainty in this analysis due to imprecise mass movement locations. This work offers new insights into the role of wildfire on mass movement susceptibility, representing a first step towards broader understanding of regional triggering mechanisms. Future efforts should incorporate additional high-accuracy mass movement locations (e.g. $\sim 500m$) that are more

representatively distributed around the globe to further advance understanding into mass movement responses across climates and regions.

*Author contributions.* Elsa Culler and Ben Livneh designed the experiments in consultation with all co-authors. Balaji Rajagopalan assisted with the design of the statistical analysis and Kristy Tiampo aided in the analysis of the mass movement triggers. Elsa Culler processed the data, developed the model code and performed the statistical analysis. Elsa Culler prepared the manuscript with contributions from all

co-authors.

*Competing interests.* The authors declare that they have no conflict of interest.

*Acknowledgements.* This research was funded by NASA IDS grant 16-IDS16-0075, The Interaction of Mass Movements with Natural Hazards Under Changing Hydrologic Conditions.

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
