# Peer review of "A data-driven evaluation of post-fire landslide susceptibility"

_Natural Hazards and Earth System Sciences, 2021_

## Referee Report (RR1)

**REVIEW**
**A data-driven evaluation of post-fire landslide susceptibility**

Elsa S. Culler, Ben Livneh, Balaji Rajagopalan, and Kristy F. Tiampo

**REMARKS FOR THE AUTHOR**

The paper focus on the differences between the occurrence of rainfall-triggered mass movements at unburned and unburned locations working at small scale (i.e. over large areas) using landslide occurrence from a global catalogue and rainfall data. Despite the authors did a consistent set of analyses, there are many uncertainties on what they found with many aspects of the analysis, results and conclusion to be clarified, modified or avoided. In many parts the authors attempt to speculate on specific process differences but without proper evidences, that probably could not be solved at this scale, unless specific process data are provided and analysed. Please find in the following the specific comments to the manuscript.

**COMMENTS AND SUGGESTION TO THE AUTHOR**

**ABSTRACT**

| | | |
|---|---|---|
| **Page 1** | **Line 1** | Landslide and debris flows are just a part of the geo-hydrological phenomena that can be impacted be fires. Please mention also the other phenomena or clarify that these are only some of them. |
| **Page 1** | **Line 4** | Please specify why GLC should facilitate regional inter-comparison? |
| **Page 1** | **Line 8** | The authors here speculate on the seasonality of "mass movement-triggering storms" but actually this should be read as the seasonality of mass movements triggered by rainfall. In addition, please specify what "other rainfall-triggered mass movements" means; here is too generic. |
| **Page 1** | **Line 12** | "… characteristics of rainfall-triggered mass movements …".in general, or only fire-related? |

**TEXT**

| | | |
|---|---|---|
| **Page 1** | **Line 15** | Please specify what the author intend for "path"? Propagation, runout path? |
| **Page 1** | **Line 24** | "sediment-laden floods" or "sediment-laden flows", since the authors use the term mass- movements, I assume they refer to the second. |

| | | |
|---|---|---|
| **Page 1** | **Line 29** | Since this is a generic/general statement should be referred not necessarily to US recent works. In addition, I suggest to avoid defining "meteorology" or "length of time since the most recent fire" as factors, since those may be just indirect ways to refer to proper landslide conditioning factors. |
| **Page 2** | **Line 36** | This sentence is really cryptic. What is the "relative magnitude of triggering precipitation events", why this should be a proxy for the susceptibility? Is this a result of the study or is an initial assumption? |
| **Page 4** | **Line 90** | Here use the term "widely recognized relationship" in place of "statistically significant positive relationship", since you are mostly referring to literature and not to specific statistical tests results. |
| **Page 4** | **Line 100** | "is a universal phenomenon" seems a bit ambitious here. |
| **Page 5** | **Line 124** | "precision" or "accuracy", here and at line 132 you are using the wo worlds but they do not refer to the same problem. In addition, completeness may not be necessarily a problem when using inferential statistics. |
| **Page 5** | **Line 145** | Here and in the rest of the text, the readers have the impression that the authors just considered part (mostly US) of the literature. I'm not a specific expert of fire related mass movements, but just a quick search on the main scientific literature search engines revealed also specific studies in other part of the world. Since the authors use this lack of local studies as one if not the main justification for the work, this problem is relevant. Hence, please account also the other studies in different countries and modify the text accordingly. |
| **Page 5** | **Line 151** | What is the rationale behind the choice of "the seven-day running total precipitation depth percentile for the 30 days surrounding the day of the year"? Which percentile do the authors refer to? Which day of the year is used? Need to clarify why these should be used as a "proxy for mass movement susceptibility". The rest of the paragraph till the beginning of section 2.1 give a series of details that are just confusing the reader. This is really cryptic, for this reason I suggest the authors to simplify this part, identifying the methodology with "understandable" steps and demanding the specificities and explanations of methods to the dedicated sections. |
| **Page 6** | **Line 162** | It will be more correct defining this a "sample" and not a "large sample". |
| **Page 6** | **Line 174** | What does "recorded locations" refer to? Fire location? |
| **Page 6** | **Line 180** | Please explain how the "hierarchical clustering algorithm" based solely on latitude and longitude is able to highlight/account for climate differences. This will be an important information to complete the description of the procedure. |
| **Page 9** | **Line 205** | Event the opposite case is possible. Please comment in the text. |
| **Page 9** | **Line 218** | What does "significantly significant differences" mean? |

**Page 10  Line 231**   It is not clear how a "7-day running average of antecedent precipitation" is able to highlight "storms of different lengths and intensities"? Please specify.

**Page 10  Line 233**   Please specify what type of "7-day antecedent rainfall indices".

**Page 10  Line 234**   "more equal comparison of mass movement triggers which fall within throughout this spectrum of storm intensity"? This is really cryptic. Please specify. In addition, "is less sensitive to small errors in precipitation" is really tautologic, since this is an aggregated measures; better removing it. It is unclear why being "less sensitive to mass movement date accuracy" is an advantage for this type of analysis. Maybe this hides important causal relations between rainfall and post-fire landslide occurrences, in line with what you have mentioned before about the importance of runoff related phenomena compared to the infiltration related once.

**Page 10  Line 239**   The sentence is not clear! Please rephrase!

**Page 11  Line 256**   How do you exactly normalized these value (i.e. which kind of calculation did you do)? Please also explain exactly why the normalization you are performing should "facilitate the comparison of mass movement-triggering events across a variety of seasons and climates". Is this based on previous study? Please also justify why "this statistic controls for geographic and seasonal differences across mass movement events". Which is the rationale behind that?

**Page 11  Line 265**   Again, here and hereafter, it's unclear why the percentile should serve as a proxy for relative mass movement susceptibility?

**Page 11  Line 270**   Mann–Whitney test does compare ranks and not median directly, or at least it may compare medians under certain circumstances and distribution assumptions (i.e. the two samples should have the same shape). Please check.

**Page 11  Line 272**   The authors do not provide any evidence on the fact that data are uniformly distributed, and is unclear how Sect 2.4 should guarantee this. In addition, Mann–Whitney test does require any uniform distribution. Instead it requires that the two analysed samples should be measured on a ordinal or continuous scale and should not normally distributed.

**Page 11  Line 280**   It is unclear what " …with the actual sample number adjusted by region so that all sites were selected evenly" mean.

**Page 11  Line 283**   What do you intend whit "lead time"? Is this the time of occurrence of the mass movement?

**Page 11  Line 283**   If I well understood you "control dataset" include those rainfall cumulative values in the selected accumulation period (i.e. the 7 day period) in a period of 15 days before and after the date of landslide. This assume a pretty constant rainfall characteristics over time and in particular over the same 30 days period across all

the years. Given the general rainfall variability I'm not sure this 30-days period length is large enough to considered all the possible climatic variability in the selected area, and I believe it should be better to consider a seasonal period to estimate the "control" references. Alternatively, the authors could show the influence of the selection of the length of the "control" period on their results. This comment is someway related to the previous comment "Page 10 Line 231", please consider them related (the larger is this accumulation period, the larger should be the control period).

**Page 11 Line 295**  What do the authors intend for seasonality? Please give a definition or refer to a reference one.

**Page 12 Line 301**  "frequency" of what?

**Page 12 Line 301**  "These persistent differences between burned and unburned sites were removed by subtracting the mean precipitation frequency for both the burned and unburned groups": is the intent of the authors to perform a variable scaling? Why don't they do a full variable standardization dividing by the standard deviation? Please explain better how this should be useful in the analysis?

**Page 13 Line 319**  From here to the end of the paragraph: so basically, you got confirmation on the effect or fires only in the area in US with previous studies, while your main hypothesis is to test this possible effect worldwide. Please comment

**Page 13 Line 326**  Figure 4 and figure 5 are only showing the p-value. Without boxplots it is impossible to check whether a p-value greater than 0.05 correspond to a precipitation percentile in burned areas lower than the unburned ones. This is a relevant information for the analyses. I suggest to realize plots or multiple plots similar to that in fig 3 or modifying the plot in a way to show such information. Please homogenize the p-value threshold descriptions in the figure captions.

**Page 17 Line 343**  I assume the figure 6 refers to the output of Mann–Whitney test even if in they axis is specified "Wilcox" test, which actually should read a "Wilcoxon". Maybe this is just a refuse since the Mann-Whitney test is also called Wilcoxon rank-sum test, but there is a completely different Wilcoxon test called "Wilcoxon signed-rank test" which test something different. Since some of the interpretation of p-value done by the authors make me think on the use of this last, I ask them to check. Indeed, it seems (i.e. al least observing plots styles) that these test have been done in R which uses the same function (i.e. *wilcox.test()* fuction) to perform different types of Wilcoxon tests. Plase highlight the meaning of the different colour tones in the caption/tex.t

**Page 17 Line 349**  "with lower values": similarly, to comment Page 13 Line 326, p-value may just highlight some difference but not their positive or negative sing. This information in figure 6 can be only appreciated in panels (h)-(u) but only for the case of the day-of-landslide precipitation.

**Page 17 Line 357** "with implications for potentially region-specific physical processes associated with mass movement triggers": the analysis of timing of rainfall/storms in different regions cannot say anything in this regard, these may only say something on the rainfall characteristics leading to landslides which presumably depends from regional climatic difference.

**Page 17 Line 358** "in the Himalayas and Southeast Asia (Fig. 6 panels (f) and (g)) precipitation rises at a similar rate for each group": It is really difficult to appreciate this from the figure! Indeed, there seems to be substantial differences between burned and unburned areas. In addition, from here to the end of the paragraph all the speculations/comments of the rainfall intensity do not find support from the analysis of the figure, or at least the authors do not provide all the information to appreciate this (only panels (h)-(u) may provide such information). For instance, why "In the Pacific Northwest and California (Fig. 6 panels (d) and (b)), the burned sites exhibit shorter but more intense storms than the unburned sites in the week preceding the mass movement"?

**Page 19 Line 371** This does not seems the case for all the panels, but is almost impossible to appreciate correctly. Please add 0.9 p-value line or at least their ticks in Figure.

**Page 19 Line 378** Fig.5 should probably read as Fig. 7.

**Page 19 Line 381** 4854%? Please check.

**Page 19 Line 384** Maybe hereafter, it will be worthwhile to mention the rainfall season. This is the really relevant information to speculate on landslide occurrence (e.g. in the Himalaya the monsoon period is generally in June and bring almost the entire yearly rainfall leading to landslides, but this is not the same for other regions characterized by different seasonal rainfall regimes and distributions). How the analysis on seasonality account this?

**Page 19 Line 388** Please see previous comments on this type of plots (Figure 4 and 5), and on their possible interpretation regarding rainfall.

**Page 19 Line 393** Why? p-value > 0.05 only for "less then 1 year"

**Page 19 Line 395** Why? The p-values are always > 0.05 for all the lines. Here the authors seem using a p-value interpretation opposite to what in Fig 4 and 5 which should be the correct one. Is something missing here for the interpretation?

**Page 19 Line 415** This is certainly visible for California, but not for Himalayas!

**Page 19 Line 427** In the view of the comments above, the discussion may be revised carefully and have significant changes. In the section some points are really too much speculative with no support from the data and results described in the manuscript: please revise these parts and indicate appropriate references to support them.

**Page 19  Line 505**  In the view of the comments above, also the conclusion may be revised carefully and have significant changes. Please maintain only the relevant findings, avoiding really speculative conclusions not well supported by data and analyses' results

---

## Referee Report (RR2)

**Referee Report**

1) The study areas are limited only in the USA, Central America, the Himalayas, and Southeast Asia. As a result, discussion and conclusions should focus on the differences of rainfall intensity and duration that exist in each area and distinguish the respective thresholds that initiate landsliding in each region.

2) The geological setting which is one of the most important predisposing landslide factors is entirely absent from the paper. For example, geological conditions and landslide mechanism is entirely different in the Himalayas and North America. Authors should comment on that. In addition, they should categorize all landslides in surficial and bedrock type landslides since rainfall intensity and duration as a trigger mechanism is completely different between these types. Furthermore, authors should explain if the data sample of n=5313 includes both landslides and rockfalls.

3) It is not clear if earthquake or snowmelt triggered landslides have been eliminated from the data sample of n=5313. Which was the original sample number, how many earthquakes or snowmelt triggered landslides were eliminated and which are the respective maps, with the sampling areas before and after the landslide elimination?

4) The data set used in the analysis has no uniformity. It should be on the same temporal range, following the range of the NASA Global Landslide Catalog 1988–2015, or 2007-2015

5) Since the authors have used MODIS Terra and Aqua satellites 2000-2020, which is the percentage of potential post-wildfire landslide events among the period 2000-2015 that coincides with the NASA Global Landslide Catalog?

6) According to the authors landslides were classified as burned if any part of the area where the mass movement occurred was burned at some point within the three years prior to the event to capture both waves of increased susceptibility without over-identifying mass movements areas where fires occur every few years. Authors should explain why they used there years prior to the event and not more or less than three years

7) According to the authors 489 mass movements (9:2%) were categorized as potential post-wildfire events. Which of these mass movements belong to landslides or rockfalls?

8) Precipitation analysis (intensity or duration) is different in different types of mass movements. Authors should clarify what types of mass movements they are focusing on.

9) Finally, it is not clear if rainstorms or rainfall intensity have triggered landslides only in the burned areas, since rainfall intensity is a common triggering mechanism in both burned and burned areas. Authors should explain this more thoroughly.

---

## Author Response (AR2)

**Response to reviewers**

RC1

This work explores the relationships between wildfires and landslide susceptibility in various regions of the world. The results outline the complexity of these relationships and have permitted to derive some conclusions on the smaller amounts of precipitation needed for landslide triggering in burned areas and on the seasonal shift in landslides occurrence. I am reporting below some suggestions for paper revision.

We address the reviewer's concerns below:

How were the study regions selected? Since the availability of data on both vegetation fires and landslides is fundamental in the choice of the study areas, one could ask why other regions where such data are available, for instance, Europe and Australia, were not considered.

We thank the reviewer for this question. The data from Europe and Australia were excluded because only a very small percentage of the landslides in these regions could be identified as recently burned. We add the following text to clarify this aspect of the study region selection:

Regions were determined using the AGglomerative NESting (AGNES) hierarchical clustering algorithm (Kaufman and Rousseeuw, 2009) considering the latitude and longitude of the landslides, and clusters were subsequently combined, split, or eliminated on the basis of sample sizes as described below. First, the cluster tree was truncated at 30 clusters, after which all the clusters with fewer than 100 data points or less than 5% burned sites were eliminated. Notably, two commonly studied regions for landslides - Europe and Australia (e.g. Van Den Eekhaut, 2020; Nyman, 2011) - were eliminated due to a lack of verifiable post-wildfire landslides available in the GLC. Cases where two nearby regions with lower numbers of landslides, for example, Central America and Caribbean/Venezuela, were joined manually. Finally, the largest region, encompassing Western US and Canada, was split into three sub-regions based on an additional identical clustering process over this sub-domain. The final regions are shown in Fig. 1panel (a). The Pacific Northwest of North America was included even though the percentage of burned sites is lower than threshold, but at 4.4% it was nearly double the highest percentage among the eliminated regions 2.25% in the Eastern US). Some landslides were not included in any of the final regions.

These events were not, however, eliminated from any analysis of all landslides.

Nyman, P., Sheridan, G. J., Smith, H. G., & Lane, P. N. (2011). Evidence of debris flow occurrence after wildfire in upland catchments of south-east Australia. *Geomorphology*, *125*(3), 383–401. https://doi.org/10.1016/j.geomorph.2010.10.016 Van Den Eeckhaut, M., & Hervás, J. (2020). State of the art of national landslide databases in Europe and their potential for assessing landslide susceptibility, hazard and risk. *Geomorphology*, *139–140*, 545–558. https://doi.org/10.1016/j.geomorph.2011.12.006

**Although this paper deals with rainfall-triggered landslides, other factors that influence the occurrence of landslides - e.g. earthquakes - could be mentioned, even if only to clarify that these factors are not relevant in the study regions and the considered years.**

We thank the reviewer for this observation. The following section clarifies that landslide sites were excluded if they were marked as related to other factors such as earthquakes or snowmelt:

To reduce errors resulting from including a variety of types of rainfalltriggered landslides within the same dataset, the selected landslides were limited to those categorized by a `landslide trigger' value of `rain,' `downpour,' `flooding,' or `continuous rain.' *Landslides with a second trigger such as an earthquake were eliminated.* Snowmelt-driven landslides were also not included because the impact of precipitation is delayed in those cases -an analysis of the snow record in California/Nevada revealed only a single event with enough antecedent snow to suggest it could have been mislabeled.

Although it focuses on a specific issue and a particular type of mass movement, the work by Riley et al. (2013) on the frequency-magnitude relationships of debris flows could be mentioned in the introduction and/or in the discussion as it compares fire-related and non-fire related debris flows at the global scale. *Riley KL, Bendick R, Hyde KD, Gabet EJ. 2013. Frequencymagnitude distribution of debris flows compiled from global data, and comparison with post-fire debris flows in the western US. Geomorphology, 191: 18–128.* https://doi.org/10.1016/j.geomorph.2013.03.008.

We thank the reviewer for this suggestion, and will include this reference in the introduction:

A study by Riley et al. (2013) comparing post-wildfire and non-fire-related debris flows on a global scale found that the volumes of the post-wildfire debris flows tended to be smaller. This finding suggests an increase in debris flow hazard and frequency after wildfires.

While it is important to acknowledge the problems in the quality of data, the possible occurrence of "many false positive burned landslides" mentioned in the discussion (page 20, lines 414-416) could partly undermine the results of this study. Saying that a validation of which landslides were truly post-wildfire is outside the scope of the study is a rather weak way to cope with this issue. The authors could try to better clarify which datasets are affected by these problems and delimit the extent and severity of these errors.

We thank the reviewer for this comment. We include additional analysis and discussion of the issue of false positive burned areas as follows. Firstly, we have computed the percentages of burned sites among landslides with differing location accuracies to quantify the potential error. Secondly, we propose to include additional analysis of the results, splitting the landslides into 'high accuracy' and 'low accuracy groups. The following will be added to the methods section to explain this analysis:

To explore the effects of variability in location accuracy and landslide type within the GLC, validation analyses were performed to quantify the extent of errors due to these factors. Firstly, the percentages of burned sites in each region were computed for each location accuracy. Subsequently, the results of the Mann-Whitney hypothesis tests comparing pre-landslide precipitation percentiles were duplicated splitting the data in the high- and low-accuracy groups (<=1 km and > 1 km respectively). The number of days with significantly significant differences in precipitation percentile in the 14 days prior to the landslide and 7 days are computed in each group.

The following additional figure and accompanying text will be included to address this issue (the figure number 3a is a placeholder so as not to confuse it with existing figures):

Figure 3b: p-values for Mann-Whitney hypothesis tests comparing precipitation percentiles at burned and unburned sites. The thick black line shows the p-values for all landslides, while green and orange lines show high (1 km or less) and low (greater than 1 km) location accuracies. A horizontal black line shows the p=0.05 significance threshold, while a vertical black line indicates the day of the landslide.

*Figure 3b shows p-values for Mann-Whitney hypothesis tests comparing precipitation percentiles for burned and unburned groups for high and low location accuracy groups*

of landslides. High accuracy indicates less than 1 km. Several regions, such as California (Fig. 3b panel (b)) show substantial differences between the high-accuracy and lowaccuracy p-values. Sample sizes of burned locations among the exact locations are low, ranging from 2 to 34 in each region, with overall only 3.7% of high-accuracy landslides classified as burned (below the threshold used to exclude regions from this study). The low percentage of burned sites may partially account for high p-values among the highaccuracy group. An additional important consideration is the likelihood of a greater number of false positive burned sites among the low-accuracy group. Notably, the percentage of identified burned sites using this method increases with the location accuracy radius – globally 12.5% of low-accuracy landslides were identified as burned in contrast with only 3.7% of high-accuracy landslides.

Finally, we will expand the discussion:

Low landslide location accuracy and lower number of burned landslides may have also contributed to the lack of conclusive results in the Pacific Northwest, Southeast Asia and Central America. The regions outside the US and Canada tended to have less accurate landslide locations. Furthermore, less accurate locations were also more likely to be marked as burned, with a threefold increase in the percentage of landslides identified as burned between high- and low-accuracy groups. This occurs because larger landslide radii were more likely to contain burned area by chance alone, and hence become `false positive' post-wildfire landslides, i.e.~landslides that occurred nearby but not coincident to a burned area. This idea is supported by the lower cumulative burned fractions within the regions outside the US and Canada (see Fig. 1 panels (c) and (d)). Though landslide accuracy in the GLC is an approximate measure, introducing the possibility of false negative unburned sites, false positive post-wildfire landslides nonetheless represent an important potential source of uncertainty in this analysis. These uncertainties introduce the possibility that some of differences in triggering precipitation percentiles between burned and unburned sites may be related to unique qualities of fire-prone areas rather than fire itself. The degree to which fires and landslides are statistically linked also contributes to the rate of false positives. Some regions may have many false positive burned landslides because there was a larger percentage of low accuracy locations, or alternatively because there was no significant increase in the probability that a landslide would occur in a burned location. Such a low posterior landslide probability given that a fire has occurred would tend to greatly increase the number of false positive burned areas by decreasing the probability that a landslide occurred in the burned section of the landslide radius, thus negating the effects of larger landslide buffers. Future studies using visible and other satellite imagery to pinpoint landslide locations and dates

could help further clarify the post-wildfire posterior landslide probability by essentially eliminating the location error.

**Caption of Fig.3: it could be specified that the grey belt corresponds to the day of landslide occurrence.**

The caption of Fig. 3 will read:

Seven-day precipitation percentile in the lead-up to landslides for all landslides in (a) and for the six individual regions labeled (b)--(g), whether classified as part of one of the regions or not. *The day of the landslide is indicated with a vertical grey column.* Days where a significant difference was found between the burned and unburned groups are indicated in bold coloring (Mann--Whitney hypothesis test, p > 0.05).

**The caption of Fig. 4 is very long and not easy to follow: I wish to suggest moving part of it to the text of the manuscript.**

The following modifications have been made to the caption and text related to Fig. 4:

Figure 4: *p*-values of Mann--Whitney hypothesis tests comparing landslidetriggering precipitation relative to 100 bootstrapped samples (*n*~100 for each sample) drawn from a 38-year precipitation record from the landslide locations. The y-axes are shown with a probit transform to expand the section of the axis where *p*-values are below 0.05 (significant at 95% confidence, shown as a dashed black line). The y-axis has also been inverted so that larger differences in precipitation (lower *p*-values) are higher on the y-axis for consistency with the percentile plots in Fig. 3. In panels (h)-(u), an example of the kernel density estimate (kde) for day-of-landslide precipitation in black separated by burned and unburned groups is compared with kdes of all bootstrapped samples in orange (burned group) or purple (unburned group).

Figure 4 highlights the increase in precipitation in the days before a landslide relative to historical amounts for that location and time of year, i.e., relative to climatology, offering a robust assessment of the landslide precipitation departure. The Mann--Whitney p-values comparing the precipitation record on each day to each of the (~100) samples are shown

in \ref{fig:bootstrap} panels (a)--(g). Landslide events have been split into burned and unburned groups (shown in orange and purple respectively) for six regions and for all landslides in the study. Bootstrapped samples were drawn from the same DOY and locations as the landslides but from a randomly selected year. In panels (a)-(g), box plots of p--values represent the degree to which the landslide-triggering precipitation differed from climatological precipitation with lower values indicating a larger difference between the two precipitation distributions.

Examples of the kernel density estimates of each bootstrap sample as compared to the precipitation on the day of the landslide are shown in Fig. 4 panels (h)--(u) to better illustrate the comparisons made by the hypothesis tests in panels (a)--(g). Each orange or purple curve was tested against the black curve to obtain the boxplots of p-values at 0 days before the landslide.

**RC2**

This manuscript pulls in several interesting global datasets to try to add more data and a global perspective to the existing literature on wildfire and landslides. Currently, there are a few relatively large challenges for the manuscript that lead to a lack of clarity, generally. I will point out several of these challenges and potential solutions that might help the authors to refine their description to enhance clarity and ultimately usability of the results.

The first challenge is that the authors do not differentiate between landslides and debris flows following wildfire. This is problematic because there is a very large body of work that exists on post-wildfire debris flows, and a smaller, but important body of work on post-fire landsliding. I would highly encourage the authors to make this distinction using terminology such as the Varnes 1978 classification. The reason this is important is because the mechanisms that generate these different types of mass movement are very different and occur at very different times following wildfire. For example, post-wildfire debris flows typically happen in the first year after a fire and they are generated by distributed overland flow that coalesces into channels and mobilizes sediment (see for example McGuire 2017 and references therein). By contrast, shallow landsliding often happens decades after fire due to soil saturation and loss of root cohesion (e.g. Jackson and Roering, 2009 and references therein). These mechanisms are nearly polar opposite, in that the first is generated by very low infiltration after fire, the second is generated during a condition of very high infiltration after fire. Lumping

these two types of mass movement together makes it extremely confusing for readers to put your precipitation analysis into the proper context. Even though debris flows and shallow landslides both move rock and sediment and involve some water, most of the erosion by debris flows happens in channels whereas most of the erosion from shallow landslides happens on hillslopes. This is sort of like saying that bread and dog biscuits are similar because they involve grain and baking, but functionally, they are extremely different. Consequently, if you could clarify what types of mass movement you are focusing on, that would go a very long way to improving the current manuscript.

We appreciate this concern. The largest category of landslides included in the NASA GLC is labeled 'landslide', and includes mass movements of all types. We acknowledge that the lack of differentiation as to the types of mass movements is of concern with this data source, and will include the following additional analysis to highlight this issue:

Subsequently, the results of the Wilcox tests comparing pre-landslide precipitation percentiles are duplicated splitting the data in the high- and low-accuracy groups (<=1 km and > 1 km respectively). The number of days with significantly significant differences in precipitation percentile in the 14 days prior to the landslide and 7 days are computed in each group. Finally, a similar analysis compared debris flows (labeled as 'debris flow' or 'mudslide' in the GLC) and other types of mass movements.

In addition, we include an additional figure and analysis as described comparing the day-of-landslide precipitation percentile from the undifferentiated 'landslide' group with landslide specifically labeled as 'mudslide' or 'debris flow':

---

## Author Response (AR3)

Referee #3

ABSTRACT

Page 1 Line 1 Landslide and debris flows are just a part of the geo-hydrological phenomena that can be impacted be fires. Please mention also the other phenomena or clarify that these are only some of them.

We have addressed this concern by changing the following text on line 1 of the original manuscript:

"Wildfires change the hydrologic and geomorphic response of watersheds, which has been associated with cascading hazards that include shallow landslides and debris flows."

To:

"Wildfires change the hydrologic and geomorphic response of watersheds, which has been associated *with cascades of additional hazards and management challenges. Among these continuing post-wildfire events are shallow landslides and debris flows.*"

We further return to this statement by the following reference to the Introduction (line 23): "Here, we focus on a particular sequence of cascading natural hazards known as the post-wildfire landslide. In these events, wildfires are followed by intense precipitation leading to mass movements *such as shallow landslides, or debris flows. The impact of wildfires, which themselves occur more frequently and severely as a consequence of higher temperatures and increasingly widespread drought, can lead to a variety of geo-hydrological hazards including increased snowmelt, water contamination, increased erosion rates, and decreased infiltration (AghaKouchack et al., 2020).* Post-wildfire landslides in particular occur when wildfires are followed by intense precipitation, leading to mass movements such as a sediment-laden floods, shallow landslides, or debris flows."

AghaKouchak, A., Chiang, F., Huning, L. S., Love, C. A., Mallakpour, I., Mazdiyasni, O., ... & Sadegh, M. (2020). Climate extremes and compound hazards in a warming world. *Annual Review of Earth and Planetary Sciences*, 48, 519-548.

Page 1 Line 4 Please specify why GLC should facilitate regional inter-comparison?

We appreciate this comment and include the following explanation:

"Landslide events are selected from the NASA Global Landslide Catalog (GLC). Since this catalog contains events from multiple regions worldwide, it allows a greater degree of interregional comparison than many highly localized catalogs."

Page 1 Line 8 The authors here speculate on the seasonality of "mass movement-triggering storms" but actually this should be read as the seasonality of mass movements triggered by rainfall. In addition, please specify what "other rainfall-triggered mass movements" means; here is too generic.

We thank the reviewer for this comment and clarify as follows:

"An analysis of the seasonality of mass movements at burned and unburned locations shows that mass movement-triggering storms in burned locations tend to exhibit different seasonality from rainfall-triggered mass movements *in areas undisturbed by recent fire.*"

Page 1 Line 12 "... characteristics of rainfall-triggered mass movements ...".in general, or only fire-related?

We clarify this text as follows:

"Overall, this manuscript offers an exploration of regional differences in the characteristics of rainfall-triggered mass movements *at burned and unburned sites* over a broad spatial scale and encompassing a variety of climates and geographies."

Page 1 Line 15 Please specify what the author intend for "path"? Propagation, runout path? We thank the reviewer for this comment, and will replace this sentence with:

"Mass movements are destructive when they occur near vulnerable areas, causing damage to or failure of buildings, utility lines, and roadways (Highland and Bobrowski, 2008). Landslide mitigation costs in the United States (US) are approximately 2 billion USD annually, with worldwide costs much higher (Schuster and Highland, 2001). There can also be indirect impacts, such as aggradation of the streambed, or the formation of landslide dams (Glade and Crozier, 2005)."

Glade, T., & Crozier, M. J. (2005). The nature of landslide hazard impact. Landslide hazard and risk, 43-74.

Page 1 Line 24 "sediment-laden floods" or "sediment-laden flows", since the authors use the term mass- movements, I assume they refer to the second.

We appreciate this point since floods would not be included in the inventory, and edit the text to:

"Here, we focus on a particular sequence of cascading natural hazards known as the post-wildfire landslide. In these events, wildfires are followed by intense precipitation leading to mass movements *such as shallow landslides, or debris flows.*"

Page 1 Line 29 Since this is a generic/general statement should be referred not necessarily to US recent works. In addition, I suggest to avoid defining "meteorology" or "length of time since the most recent fire" as factors, since those may be just indirect ways to refer to proper landslide conditioning factors. AND Page 2 Line 36 This sentence is really cryptic. What is the "relative magnitude of triggering precipitation events", why this should be a proxy for the susceptibility? Is this a result of the study or is an initial assumption?

We have reconfigured these last two paragraphs of section 1 to address the two concerns above:

"There are numerous local studies demonstrating a relationship between wildfire occurrence or severity and the amount of precipitation that triggers a mass movement (Cannon et al., 2008; Gartner, 2005; Reneau et al., 2007; Riley et al., 2013). The impact of wildfire on landslide hazards can also vary on the basis of local factors such as vegetation, and soil type (Cannon et al., 2010; Staley et al., 2018). In general, the lack of complete landslide inventories including a wide variety of climates and ecoregions presents an obstacle to an evaluation of the role of fire in rainfall-triggered landslides, and regional studies are exceptional (Klose, 2015b).

This study seeks to test the hypothesis that wildfire consistently increases mass movement susceptibility across six global regions. In this large sample of mass movement events (n=5313), higher expected frequency of precipitation is indicative of greater mass movement susceptibility. Though we cannot draw conclusions about the susceptibility leading to any particular event, higher median expected frequency suggests that the threshold for triggering a mass movement was lowered, e.g. susceptibility was greater. A second purpose of this study is to explore the possibility that the relationship between wildfire history and the expected frequency of landslide-triggering precipitation varies by regions."

Page 4 Line 90 Here use the term "widely recognized relationship" in place of "statistically significant positive relationship", since you are mostly referring to literature and not to specific statistical tests results.

We appreciate this suggestion and have made the following change to the text: "The *widely recognized* relationship between mass movements and burn severity suggests that mass movement susceptibility increases after wildfires in the Western US,"

Page 4 Line 100 "is a universal phenomenon" seems a bit ambitious here.

We modify this sentence to:

"A global study by Riley et al. (2013) comparing post-wildfire a non-fire-related debris flows found that the volumes of the post-wildfire debris flows tended to be smaller. This finding suggests that the increase in debris flow hazard and frequency after wildfires *occurs in a variety of environments.*"

Page 5 Line 124 "precision" or "accuracy", here and at line 132 you are using the wo worlds but they do not refer to the same problem. In addition, completeness may not be necessarily a problem when using inferential statistics.

We thank the reviewer for this comment and adjust the text as follows:

For this study, we chose to use the NASA Global Landslide Catalog (GLC, Kirschbaum et al., 2010). As with the few other regional and global databases available, the broad domain of the GLC comes coupled with issues of *location error* and spatial bias. For each landslide location, the GLC contains an estimate of the area in which the landslide occurred, labeled the "location accuracy". For consistency, we refer to this parameter using the same name.

. . .

Despite limitations, the GLC was chosen for this study primarily because as of this writing it offers the largest spatial range of any catalog.

Page 5 Line 145 Here and in the rest of the text, the readers have the impression that the authors just considered part (mostly US) of the literature. I'm not a specific expert of fire related mass movements, but just a quick search on the main scientific literature search engines revealed also specific studies in other part of the world. Since the authors use this lack of local studies as one if

not the main justification for the work, this problem is relevant. Hence, please account also the other studies in different countries and modify the text accordingly.

Finally, in contrast to post-wildfire mass movement studies focused on a specific regions like the western US (Cannon and DeGraff, 2009), southern California (Gartner et al., 2014), *Western Canada (Jordan, 2014), Korea (Jong-Ook et al., 2018)* or southeast Australia (Nyman et al., 2011), this study combines the GLC with globally-observed fire and precipitation data to offer unique insights into the role of fire on mass movement susceptibility in diverse regions across the globe.

Jordan Peter (2016) Post-wildfire debris flows in southern British Columbia, Canada. *International Journal of Wildland Fire* **25**, 322-336.

Lee, J.-O., Lee, D.-K., & Song, Y.-I. (2019). Analysis of the potential landslide hazard after wildfire considering compound disaster effect. *Journal of the Korea Society of Environmental Restoration Technology*, 22(1), 33–45. https://doi.org/10.13087/KOSERT.2019.22.1.33

Page 5 Line 151 What is the rationale behind the choice of "the seven-day running total precipitation depth percentile for the 30 days surrounding the day of the year"? Which percentile do the authors refer to? Which day of the year is used? Need to clarify why these should be used as a "proxy for mass movement susceptibility". The rest of the paragraph till the beginning of section 2.1 give a series of details that are just confusing the reader. This is really cryptic, for this reason I suggest the authors to simplify this part, identifying the methodology with "understandable" steps and demanding the specificities and explanations of methods to the dedicated sections.

First, the seven-day running total precipitation depth percentile for the 30 days surrounding the day of the year and across the total 38-year record (see Sect. 2.4) was used as a proxy for mass movement susceptibility. We assume here that greater susceptibility results in a lower precipitation threshold to trigger a landslide. An observation, therefore, of lower precipitation percentile values triggering storms across a sample of sites suggests that susceptibility is generally higher in that group. This principle is illustrated in the susceptibility-based rainfall threshold model developed by Monsieurs et al. (2019), in which the predicted threshold of antecedent rainfall resulting in a landslide is adjusted according to susceptibility factors.

Monsieurs, E., Dewitte, O., & Demoulin, A. (2019). A susceptibility-based rainfall threshold approach for landslide occurrence. *Natural Hazards and Earth System Sciences*, 19(4), 775-789.

- Page 6 Line 162 It will be more correct defining this a "sample" and not a "large sample". This change has been made
- Page 6 Line 174 What does "recorded locations" refer to? Fire location? We have clarified this text as follows: Only records with location uncertainty of 10 km or less were included,

Page 6 Line 180 Please explain how the "hierarchical clustering algorithm" based solely on latitude and longitude is able to highlight/account for climate differences. This will be an important information to complete the description of the procedure.

To compare the differences in mass movement triggers in different climates, we divided the mass movements into regions (see Fig. 1 panels (a) and (b)). Regions were determined using the AGglomerative NESting (AGNES) hierarchical clustering algorithm (Kaufman and Rousseeuw, 2009) considering the latitude and longitude of the mass movements, and clusters were subsequently combined, split, or eliminated on the basis of equalizing sample sizes as described below. *Though the regions are still large enough to encompass considerable variability in climate, the spatial clustering helps to ensure that the variability across regions – particularly in latitude - is larger than the variability within.*

Page 9 Line 205 Event the opposite case is possible. Please comment in the text.

We thank the reviewer for this observation and correct the text as follows:

Due to uncertainty in the locations of many of the mass movement locations, both false positive and false negative errors in burn history classification are possible. Some mass movements classified as burned may have occurred near a recent fire but not within the fire perimeter, or conversely some mass movements classified as unburned may in fact have been located inside a fire perimeter but near the edge. However, by classifying mass movements as burned if any part of the potential location was burned limits the potential for false negative errors while increasing the possibility of false positive errors. For this reason we refer to mass movements as 'burned' instead of post-wildfire in this analysis. Also important to note is that false positive burned classification is a function of both the burned fraction and the conditional probability of mass movement occurrence given that a fire has occurred.

Page 9 Line 218 What does "significantly significant differences" mean?

We appreciate this observation and have corrected the text as below:

The number of days with *statistically* significant differences in precipitation percentile in the 14 days prior to the mass movement and 7 days are computed in each group.

Page 10 Line 231 It is not clear how a "7-day running average of antecedent precipitation" is able to highlight "storms of different lengths and intensities"? Please specify.

"Mass movements can be triggered by intense and short storms, by long storms of lower intensity, or somewhere in-between. A seven-day running sum of antecedent precipitation was computed to enable direct comparison of the mass movements triggered by storms a range of durations."

Page 10 Line 233 Please specify what type of "7-day antecedent rainfall indices".

We clarify the text below and provide additional support for this choice:

While including an estimate of the soil moisture was outside the scope of this study, 7-day antecedent rainfall indices *consisting of a weighted average of precipitation over the 7-day time period* have been used by other modelling studies as a surrogate for soil moisture in a combined indicator of landslide susceptibility (James and Roulet, 2009; Kirschbaum and Stanley, 2018). Furthermore, 7-day sums of precipitation have been found to perform better than other durations in threshold models of landslide occurrence (Krcak et al., 2017; Garcia-Urquia et al., 2015).

Garcia-Urquia, E., & Axelsson, K. (2015). Rainfall thresholds for the occurrence of urban landslides in Tegucigalpa, Honduras: an application of the critical rainfall intensity. *Geografiska Annaler: Series A, Physical Geography*, 97(1), 61-83. Krkač, M., Špoljarić, D., Bernat, S., & Arbanas, S. M. (2017). Method for prediction of landslide movements based on random forests. *Landslides*, 14(3), 947-960.

Page 10 Line 234 "more equal comparison of mass movement triggers which fall within throughout this spectrum of storm intensity"? This is really cryptic. Please specify. In addition, "is less sensitive to small errors in precipitation" is really tautologic, since this is an aggregated measures; better removing it. It is unclear why being "less sensitive to mass movement date accuracy" is an advantage for this type of analysis. Maybe this hides important causal relations between rainfall and post-fire landslide occurrences, in line with what you have mentioned before about the importance of runoff related phenomena compared to the infiltration related once.

We have removed this sentence in favor of the explanation above.

Page 10 Line 239 The sentence is not clear! Please rephrase!

We clarify below:

"Upon computing the CHIRPS precipitation measurements for each event, we noted that some of the ostensibly rainfall-triggered mass movements had no antecedent precipitation in the 7-day window. We removed such mass movements from the analysis."

Page 11 Line 256 How do you exactly normalized these value (i.e. which kind of calculation did you do)? Please also explain exactly why the normalization you are performing should "facilitate the comparison of mass movement-triggering events across a variety of seasons and climates". Is this based on previous study? Please also justify why "this statistic controls for geographic and seasonal differences across mass movement events". Which is the rationale behind that?

In an initial analysis of the precipitation data, we were unable to distinguish between normal seasonal increases in precipitation and the mass movement-triggering precipitation. In order to isolate that triggering storm, it was important to normalize for both location and time of year. We accomplished this by computing a 30-day rolling percentile of the 7-day running precipitation values based on 38 years of historical precipitation climatology from 1981–2019 for each location. Percentiles have been used to compare landslide-triggering precipitation across larger, e.g. country-sized regions (Kirschbaum et al., 2020 and Araujo et al., 2022) in order to control for differences in climate or precipitation data source. For this study, the percentile produced a uniform distribution of precipitation ranging from 0 to 1, controlling for geographic and seasonal differences.

Araújo, J.R., Ramos, A.M., Soares, P.M.M. *et al.* Impact of extreme rainfall events on landslide activity in Portugal under climate change scenarios. *Landslides* **19**, 2279–2293 (2022). https://doi.org/10.1007/s10346-022-01895-7

Kirschbaum, D., Kapnick, S. B., Stanley, T., & Pascale, S. (2020). Changes in extreme precipitation and landslides over High Mountain Asia. *Geophysical Research Letters*, 47, e2019GL085347. https://doi.org/10.1029/2019GL085347

Page 11 Line 265 Again, here and hereafter, it's unclear why the percentile should serve as a proxy for relative mass movement susceptibility?

Please see the additional explanation above

Page 11 Line 270 Mann–Whitney test does compare ranks and not median directly, or at least it may compare medians under certain circumstances and distribution assumptions (i.e. the two samples should have the same shape). Please check.

Please see the correction below:

"The null hypothesis of the Mann–Whitney test was that the distribution of precipitation percentile of the burned sites is generally greater than or equal to the distribution of precipitation percentiles of the unburned sites. (Helsel et al., 2020)"

Helsel, D.R., Hirsch, R.M., Ryberg, K.R., Archfield, S.A., and Gilroy, E.J., 2020, Statistical methods in water resources: U.S. Geological Survey Techniques and Methods, book 4, chap. A3, 458 p., https://doi.org/10.3133/tm4a3. [Supersedes USGS Techniques of Water-Resources Investigations, book 4, chap. A3, version 1.1.]

Page 11 Line 272 The authors do not provide any evidence on the fact that data are uniformly distributed, and is unclear how Sect 2.4 should guarantee this. In addition, Mann–Whitney test does require any uniform distribution. Instead it requires that the two analysed samples should be measured on a ordinal or continuous scale and should not normally distributed.

See the clarification below:

Percentiles are by definition uniformly distributed rather than normally distributed, making the Mann–Whitney test, which does not require normal distribution, the most appropriate hypothesis test for these data.

Page 11 Line 280 It is unclear what " ... with the actual sample number adjusted by region so that all sites were selected evenly" mean.

We clarify the text below:

(n= the smallest number above 100 that ensured each site was included in the same number of samples).

Page 11 Line 283 What do you intend whit "lead time"? Is this the time of occurrence of the mass movement?

See the clarification below:

These samples are representative of precipitation for a particular *number of days before* the mass movement and serve as a control dataset with which to compare the prelandslide precipitation.

Page 11 Line 283 If I well understood you "control dataset" include those rainfall cumulative values in the selected accumulation period (i.e. the 7 day period) in a period of 15 days before and after the date of landslide. This assume a pretty constant rainfall characteristics over time and in particular over the same 30 days period across all the years. Given the general rainfall variability I'm not sure this 30-days period length is large enough to considered all the possible climatic variability in the selected area, and I believe it should be better to consider a seasonal period to estimate the "control" references. Alternatively, the authors could show the influence of the selection of the length of the "control" period on their results. This comment is someway related to the previous comment "Page 10 Line 231", please consider them related (the larger is this accumulation period, the larger should be the control period).

We appreciate this concern. The choice of 30 days was motivated by

- a) preliminary analysis in which it was difficult to distinguish landslide-triggering precipitation from ordinary seasonal changes. For example, precipitation typically increases substantially over time during the fall months in California, and so a seasonal percentile would result in a typical but misleading amount of increase in precipitation percentile that
- b) Climatology data are frequently monthly

Page 11 Line 295 What do the authors intend for seasonality? Please give a definition or refer to a reference one.

**2.7. Mass movement seasonality experiment**

The probability of landslide occurrence in a given temporospatial domain varies throughout the year (Stanley et al., 2020); we refer to this annual pattern for a given domain as mass movement seasonality. We bypothesize that wildfire alters mass movement seasonality. To test this hypothesis, we estimated precipitation frequency at the mass movement sites over time by computing the fraction of sites in the burned and unburned groups that had precipitation on any given day.

Stanley, T. A., Kirschbaum, D. B., Sobieszczyk, S., Jasinski, M. F., Borak, J. S., & Slaughter, S. L. (2020). Building a landslide hazard indicator with machine learning and land surface models. *Environmental Modelling & Software*, *129*, 104692.

**Page 12 Line 301 "frequency" of what?**

We found that in most regions there was a long-term difference in the mean annual *precipitation* frequency

Page 12 Line 301 "These persistent differences between burned and unburned sites were removed by subtracting the mean precipitation frequency for both the burned and unburned groups": is the intent of the authors to perform a variable scaling? Why don't they do a full variable standardization dividing by the standard deviation? Please explain better how this should be useful in the analysis?

We thank the reviewer for this suggestion, and have revised the analysis to standardize the data:

These persistent differences between burned and unburned sites were removed by standardizing the mean precipitation frequency for both the burned and unburned groups, that is to say subtracting the mean and frequency and dividing by the standard deviation.